



# Insights into HO$_X$ and RO$_X$ chemistry in the boreal forest via measurement of peroxyacetic acid, peroxyacetic nitric anhydride (PAN) and hydrogen peroxide

John N. Crowley[1], Nicolas Pouvesle[1], Gavin J. Phillips[1], Raoul Axinte[1], Horst Fischer[1], Tuukka Petäjä[2], Anke Nölscher[1], Jonathan Williams[1], Korbinian Hens[1], Hartwig Harder,[1] Monica Martinez-Harder[1], Anna Novelli[1], Dagmar Kubistin[1], Birger Bohn[3], and Jos Lelieveld[1].

[1]**Division of Atmospheric Chemistry, Max-Planck-Institute für Chemie, Mainz, Germany**
[2]**Institute for Atmospheric and Earth System Research INAR / Physics, University of Helsinki, Finland**
[3]**Forschungszentrum Juelich GmbH, 52425 Juelich, Germany**

*Correspondence to*: John N. Crowley (john.crowley@mpic.de)

**Abstract.** Unlike many oxidised atmospheric trace gases, which have numerous production pathways, peroxyacetic acid (PAA) and PAN are formed almost exclusively in gas-phase reactions involving the hydroperoxy radical (HO$_2$), the acetyl peroxy radical (CH$_3$C(O)O$_2$) and NO$_2$ and are not believed to be directly emitted in significant amounts by vegetation. As the self-reaction of HO$_2$ is the main photochemical route to hydrogen peroxide (H$_2$O$_2$), simultaneous observation of PAA, PAN and H$_2$O$_2$ can provide insight into the HO$_2$ budget. We present an analysis of observations taken during a summertime campaign in a boreal forest that, in addition to natural conditions, was temporarily impacted by two biomass burning plumes. The observations were analysed using an expression based on a steady-state assumption using relative PAA-to-PAN mixing ratios to derive HO$_2$ concentrations. The steady-state approach generated HO$_2$ concentrations that were generally in reasonable agreement with measurements but sometimes overestimated those observed by factors of two or more. We also used a chemically simple, constrained box-model to analyse the formation and reaction of radicals that define the observed mixing ratios of PAA, H$_2$O$_2$. After nudging the simulation towards observations by adding extra, photochemical sources of HO$_2$ and CH$_3$C(O)O$_2$, the box model replicated the observations of PAA, H$_2$O$_2$, ROOH and OH throughout the campaign, including the biomass-burning influenced episodes during which significantly higher levels of many oxidized trace gases were observed. The model indicates that organic peroxy radicals were present at night in high concentrations that sometimes exceeded those predicted for daytime. A dominant fraction of CH$_3$O$_2$ radical generation was found to arise via reactions of the CH$_3$C(O)O$_2$ radical. Initially divergent measured and modelled HO$_2$ concentrations and daily concentration profiles are reconciled when these organic peroxy radicals are detected (as HO$_2$) at an efficiency of 35 %. The organic peroxy radicals are found to play an important role in the recycling of OH radicals subsequent to their loss via reactions with volatile organic compounds.



## 1 Introduction

Peroxyacetyl nitric anhydride ($CH_3C(O)O_2NO_2$), commonly and hereafter referred to as PAN), plays a centrally important role as a reservoir of reactive nitrogen and transportation medium for $NO_X$ from polluted to $NO_X$-poor regions of the atmosphere and thus impacts global tropospheric $O_3$ formation (Singh and Hanst, 1981; Fairlie et al., 2007; Zhang et al.,

2008). This, combined with its influence on ecosystem health and productivity (Sparks et al., 2003) have made PAN a target of environmental research for several decades (Singh, 1987; Roberts, 1990; Grosjean, 2003). PAN is formed exclusively in the termolecular reaction of $NO_2$ with the peroxyacetyl radical (PA, $CH_3C(O)O_2$), which is considered one of the four most abundant organic peroxy radicals in the atmosphere (Tyndall et al., 2001).

PAN is thermally unstable, with a lifetime for re-dissociation to reactants (R-1) which is of the order of hours at

temperatures close to 20 °C, but which increases to weeks or longer at lower temperatures as found e.g. at higher altitudes.

$$CH_3C(O)O_2 + NO_2 + M \qquad \rightarrow CH_3C(O)O_2NO_2 + M \tag{R1}$$

$$CH_3C(O)O_2NO_2 + M \qquad \rightarrow CH_3C(O)O_2 + NO_2 + M \tag{R-1}$$

As PAN is formed in reactions involving $NO_2$ and radicals formed from oxidation of organics its occurrence is frequently associated with photochemical ozone formation and PAN measurements have been interpreted to derive regional $O_3$

formation rates (Williams et al., 1997).

Peroxyacetic acid ($CH_3C(O)OOH$, hereafter PAA) is formed in a branch of the reaction of $CH_3C(O)O_2$ with $HO_2$ (R2) and is thus linked to PAN via their common organic, peroxy precursor radical.

$$CH_3C(O)O_2 + HO_2 \qquad \rightarrow CH_3C(O)OOH + O_2 \tag{R2a}$$

$$\rightarrow CH_3C(O)OH + O_3 \tag{R2b}$$

$$\rightarrow OH + CH_3CO_2 + O_2 \tag{R2c}$$

The fate of the $CH_3CO_2$ radical formed in (R2c) is decomposition to $CO_2$ and $CH_3$, the latter being converted immediately to $CH_3O_2$ in the presence of $O_2$.

Several recent experimental studies of reaction (R2) (Hasson et al., 2004; Dillon and Crowley, 2008; Jenkin et al., 2010; Groß et al., 2014) have shown that the dominant pathway (R2c) results in OH formation, contributing to $HO_X$ recycling in

$NO_X$ poor regions (Lelieveld et al., 2008; Taraborrelli et al., 2012). The IUPAC preferred values for this reaction (Atkinson et al., 2006; IUPAC, 2018) are an overall rate coefficient at room temperature of $(2.2 \pm 0.4) \times 10^{-11}$ cm$^3$ molecule$^{-1}$ s$^{-1}$ with 50% ($\pm$ 10%) of reactive collisions resulting in OH formation (R2c) and branching ratios of $0.37 \pm 0.1$ and $0.13 \pm 0.1$ for R2a and R2b, respectively.

Under conditions of temperature and pressure found in the lowermost troposphere, the rate coefficients for reaction of

$CH_3C(O)O_2$ with $NO_2$ and $HO_2$ are similar ($k_1$ and $k_2$ at 298 K and 1 bar pressure are 9.3 x $10^{-12}$ and $1.4 \times 10^{-11}$ cm$^3$ molecule$^{-1}$ s$^{-1}$, respectively) and the relative flux of $CH_3C(O)O_2$ radicals into PAN and PAA formation will depend on the relative abundance of $NO_2$ and $HO_2$. Apart from extremely clean environments or very aged air pollution, when $HO_2$ concentrations approach those of $NO_2$, this will generally favour PAN. In the summertime boundary layer at mid-latitudes,



PAN is however short lived and will readily decompose back to $NO_2 + CH_3C(O)O_2$, implying that the formation of the thermally stable peroxy-acid in (R2a) will be a significant $CH_3C(O)O_2$ and $HO_X$ sink in warm conditions with low $NO_X$ levels.

Unlike PAN, there are few measurements of PAA (Fels and Junkermann, 1994; He et al., 2010; Zhang et al., 2010) and even fewer datasets in which PAN and PAA were both monitored (Zhang et al., 2010; Phillips et al., 2013). A significant difference between PAA and many other organic acids is that, to the best of our knowledge, the former is generated in the gas-phase almost exclusively via reaction (R2a), whereas non-peroxy acids (e.g. its acetic acid co-product in R2b) may be emitted by vegetation (Talbot et al., 1995) or formed in reactions of $O_3$ with olefins (Grosjean, 1992) or by biomass burning (Talbot et al., 1988). PAA may also be formed in aerosols by the aqueous-phase oxidation of acetic acid by $H_2O_2$, but its high solubility and aqueous-phase equilibrium with $CH_3C(O)OH$ and $H_2O$ preclude significant release into the gas-phase.

Recently, we presented a dataset of quasi-simultaneous PAN and PAA measurements made at a boreal forest site in Finland (Phillips et al., 2013). In Phillips et al. (2013) more technical aspects of the measurement of PAN and PAA were described, though we also alluded to the fact that, due to their partly-common generation mechanism, PAN/PAA ratios may be a useful indicator of $HO_2$ levels. Here, we examine that aspect in more detail using an analytical expression that describes the PAA-to-PAN ratio. We also combine the PAN and PAA measurements with those of several other trace gases and use chemical box-modelling with a highly simplified reaction scheme to gain insight into $HO_X$ chemistry and the factors affecting PAN, PAA and $H_2O_2$ formation at the boreal site.

## 2 Campaign Site and Instruments

The HUMPPA-COPEC campaign in the summer of 2010 was located in the Finnish boreal forest at the SMEAR II-Hyytiälä station (Hari and Kulmala, 2005) (Latitude 61º51´N; Longitude 24º17´E). The location of the site means that it experiences a very homogeneous fetch extending over hundreds of kilometres in all directions. A campaign overview with a list of instruments and an outline of the meteorological situation during the intensive (July and August) is given in Williams et al. (2011). The campaign period was unusually warm for this location (maximum temperature recorded was ~ 30 °C), mainly due to the above average contribution of air masses from the south, which resulted in enhanced biogenic emissions from the forest and which also brought two episodes of biomass-burning impacted air to the site. The forest is dominated by a mixture of coniferous forest (Scots pine and Norway spruce) and mixed forest (conifers and Silver birch). Most of the instruments from which data has been used in this work (PAN, PAA, $H_2O_2$, NO, $NO_2$, $O_3$ and HCHO) had inlets at the top of a ~20m high tower located in a small clearing (~20m diameter) in the forest and at approximately the same height as the surrounding tree-tops. Other data including OH and $HO_2$ and organic trace gases were taken by instruments located nearby (for details of instrument positions and inlet heights see Williams et al 2011).

PAN and PAA were measured using iodide chemical ionisation mass spectrometer (I-CIMS) as described in Phillips et al. (2013). Details of the instruments used to measure $H_2O_2$ and ROOH (enzyme / fluorescence), HCHO (Hantzch method),





$NO_X$ (chemiluminescence detector) organic peroxides and $O_3$ (UV) have been reported elsewhere (Stickler et al., 2006; Hosaynali Beygi et al., 2011; Fischer et al., 2015). Details of the OH-reactivity measurements and proton-transfer Mass Spectrometric (PTRMS) / GC measurements of organic trace gases have been previously given (Nölscher et al., 2012; Yassaa et al., 2012; Kourtchev et al., 2016). OH was measured by chemical ionisation mass spectrometry (Petäjä et al.,

2009), $HO_2$ radicals were measured by laser induced fluorescence after conversion to OH as described in Hens et al. (2014) and Novelli et al. (2014). J-values were obtained from process specific parameterizations based on J-$NO_2$ and J-(O($^1$D)) measured by filter radiometers (Bohn et al., 2008). Owing to known interferences by organic peroxy radicals (Fuchs et al., 2011; Fuchs et al., 2016; Lew et al., 2018), especially in forested regions, we refer to the measurements of $HO_2$ as "LIF-$HO_2$" which represent an upper limit to true $HO_2$ levels. Since discovery of this interference in 2010, it was eliminated from

all later LIF measurements by reducing the amount of NO added internally to the sampled air and by regularly performing titration tests in the field.

## 3 Results and Discussion

Figure 1 displays measurements of PAN and PAA made by iodide-CIMS during HUMPPA-COPEC 2010 and several other trace gases ($O_3$, CO, $H_2O_2$, HCHO, $NO_X$) which provide some indication of the type of air masses sampled. Generally, the

mixing ratios of PAN and PAA were similar, with PAN typically present at less than one ppb, reflecting the relatively low $NO_X$ levels (median, noon-time mixing ratio of $0.3 \pm 0.1$ ppbv) at this boreal forest site. In two campaign periods, day 207-211 and 219-221 highlighted in grey in Figure 1, elevated levels of PAN and especially PAA were observed, with mixing ratios of PAA exceeding 1 ppb compared to levels of less than 200 pptv during the rest of the campaign. These episodes of high PAA (and PAN) were accompanied by elevated mixing ratios of longer lived trace gases such as $CH_3CN$, CO, $SO_2$,

HCHO and $O_3$, and were also coincident with higher than average temperatures. As discussed in Williams et al. (2011), $CH_3CN$ measurements indicate that during these periods, the site was impacted by biomass burning in Russia, with back trajectories suggesting that the air from the fires had travelled for a few days before reaching the site.

An additional feature during the first episode (day 210.5) is a spike in the $SO_2$, HCHO and CO mixing ratios and also in non-oxidised hydrocarbons such as pentane (Williams et al., 2011). This shorter lived plume was associated with a continuous

and rapid change in wind-direction (from ~25 to ~160 °) and higher local wind speeds (up to 25 km/hour), which resulted in arrival of biomass-burning impacted but less aged air masses to the site. The presence of sharply elevated $SO_2$ and pentane strongly suggests that fossil-fuel related emissions from St. Petersburg, (120°, 400 km distant), were mixed into this air mass, which may be regarded as chemically distinct from the other, longer-lasting biomass-burning episodes. $NO_X$ levels were not elevated during the biomass-burning plumes, confirming that they were chemically aged with respect to the

conversion of $NO_X$ to $NO_Y$. Significant increases in signals at several PTRMS masses also indicated the presence of elevated amounts of oxygenated, volatile organic compounds (OVOCs, including organic acids, aldehydes and ketones) and also aromatic trace gases during the period where the site was impacted by the biomass burning plumes. The warmer



temperatures were also associated with increases in the emissions of biogenic trace gases(Paasonen et al., 2013; Kourtchev et al., 2016), making this a chemically very complex period during the campaign. An overview of several trace gases as measured by the PTRMS is given in Figure S1 of the Supplementary Information.

Whilst the elevated levels of long lived biomass burning tracers such as CO and $CH_3CN$ and biomass burning aerosol signal (Corrigan et al., 2013) are clearly due to long-range transport, this is unlikely to be the case for the shorter-lived trace gases. The very high OH-reactivity (up to 45 $s^{-1}$) observed (Nölscher et al., 2012) during the biomass-burning events and the fact that both $H_2O_2$ and PAA display diurnal profiles consistent with photochemical generation, suggest that the high levels of PAA, PAN and $H_2O_2$ are related to the presence of high levels of radical precursors in the biomass burning plumes, as

previously observed for boreal fires (Alvarado et al., 2010).

While PAA and PAN have similar photochemical generation routes, a cursory examination of their average diel profiles during the campaign (Figure 2) reveals important differences, with peak concentrations in the daily cycle of PAA displaced (later) relative to those of PAN by about 2 hours. This is related to the maximum loss rates of PAN (via thermal decomposition) which occur at the highest temperatures in the mid-afternoon. The day-to-day variations in the diel cycle of

PAA are mainly changes in the maximum mixing ratio, reflecting variable rates of production from precursors, with a night-time concentration generally tending to zero. This is very similar to the diel profile of $H_2O_2$ which is also highly regular in shape and which, on most nights, also tends towards zero.

The mixing ratios of PAA are plotted against those of both $H_2O_2$ and PAN in Figure 3, which highlights the good correlation ($R^2 = 0.74$) with $H_2O_2$. The slope of the PAN to PAA ratio is reduced at high PAA mixing ratios, which were generally

observed when temperature and photochemical activity were highest. The more rapid loss of PAN at high temperatures causes the non-linear relationship and weakens the overall PAA to PAN correlation ($R^2 = 0.47$). The lower panel of Figure 3 indicates that PAA is strongly correlated with total organic peroxides (ROOH) and represents a significant fraction of total organic peroxides during the HUMPPA-COPEC campaign (Phillips et al., 2013). The PAN mixing ratios over the diel cycle were very variable, especially at nighttime.

Radiosondes suggest that for HUMPPA-COPEC-2010 the lowermost layer will be mixed with the residual layer by ~ 10:00 local time (07:00 UTC), and we can consider the boundary layer thereafter to be mixed up to 1 km height with a further increase in BL height over the next two hours, depending on meteorological conditions (Ouwersloot et al., 2012). As rates of entrainment from the free troposphere and the vertical gradients of e.g. PAA are not known, in the subsequent analysis we assume that photochemical generation dominates the production side of the PAA budget but re-address this later when

comparing measured and model predictions of PAA and $H_2O_2$ mixing ratios.

Analysis of the net loss of PAN, $O_3$ and $H_2O_2$ during the pre-dawn period (midnight to 4 am, see Figure 2) by fitting exponential decay curves to the data during this period gives loss rate constants of $5.2 \times 10^{-5}$ $s^{-1}$ ($H_2O_2$), $2.3 \times 10^{-5}$ $s^{-1}$ (PAN) and $9.3 \times 10^{-6}$ $s^{-1}$ ($O_3$) resulting in campaign averaged, night-time lifetimes of $\approx$ 7 hrs, 5hrs and 30 hrs, for $H_2O_2$, PAN and $O_3$, respectively. Note that as NO and $HO_2$ are absent or present in very low concentrations at night, the thermal





decomposition of PAN does not result in its loss (see later). The night-time loss rates of $H_2O_2$, PAN and $O_3$ therefore reflect different rates of deposition to the canopy, although that of $H_2O_2$ will be a lower limit as nighttime production can occur via the ozonolysis of terpenes (see below).

The average, night-time loss rate constant for PAN of $2.3 \times 10^{-5}$ s$^{-1}$ can be equated to $2V_{ex}/h$ where $V_{ex}$ is the exchange

velocity and $h$ is the boundary layer height (Shepson et al., 1992). This results in an approximate value of $V_{ex} \approx 0.23$ cms$^{-1}$ for an average, night-time boundary layer height of 200 m for this campaign (Ouwersloot et al., 2012). This is consistent with values between 0 and 0.6 cm s$^{-1}$ obtained at night-time over coniferous forests (Shepson et al., 1992; Turnipseed et al., 2006; Wolfe et al., 2009). Wolfe et al. (2009) determined that the exchange velocity of PAN increased by a factor of $\approx 4$ at noon compared to during the night, which is partially due to more efficient stomatal transport during the day. From this ratio,

yielding an average, noon-time exchange velocity of $\approx 0.9$ cm s$^{-1}$ for HUMPPA-COPEC, and assuming a daytime boundary layer height of 1400m (Ouwersloot et al., 2012), we calculate noon-time loss rate constants for PAN deposition of $\approx 1 \times 10^{-5}$ s$^{-1}$. The high temperatures encountered during the summer in HUMPPA-COPEC-2010 campaign mean that daytime loss of PAN due to deposition is small compared to thermal dissociation followed by reaction of $CH_3C(O)O_2$ with NO or $HO_2$ (see below).

With a loss rate constant of $9.3 \times 10^{-6}$ s$^{-1}$, the night-time depletion of $O_3$ is slower than for PAN. From the values presented in Figure 2, the relative loss rate of $O_3$ and PAN is 0.4, which is similar to the value of $0.42 \pm 0.19$ reported e.g. by Shepson et al. (1992) and Wolfe et al. (2009), though such comparisons are complicated by variability in the chemical losses of $O_3$ via e.g. reaction with NO or sesquiterpenes (Kurpius and Goldstein, 2003) which depend on e.g. forest type, and emission rates.

The observed (net) loss rate constant for $H_2O_2$ is $5.2 \times 10^{-5}$ s$^{-1}$, whereby $\approx 0.25$ ppbv are lost in 4 hours between 00:00 and

04:00. The change in $[H_2O_2]$ results from a combination of its production via ozonolysis of terpenes and depositional losses. During the period from midnight to 04:00, the campaign average terpene and $O_3$ mixing ratios were 300 pptv and 38 ppbv, respectively. The five biogenic species measured by the GC (isoprene, α-pinene, myrcene, 3-carene and β-pinene) have rate constants for reaction with $O_3$ that vary between $\approx 1.3 \times 10^{-17}$ to $5 \times 10^{-16}$ cm$^3$ molecule$^{-1}$ s$^{-1}$. Taking into account the mean relative concentration and $O_3$-rate coefficient for each terpene, we derive an effective, campaign averaged rate constant of

$1.5 \times 10^{-16}$ cm$^3$ molecule$^{-1}$ s$^{-1}$ for the ozonolysis of terpenes. Taking a $H_2O_2$ yield of 0.16 (e.g. as observed for α-pinene) we calculate that $\approx 0.1$ ppb $H_2O_2$ was generated during this period. Taken together with the observed (net) loss of $H_2O_2$, this implies a loss of 0.35 ppbv $H_2O_2$ by dry deposition in 4 hours, which translates to a loss rate constant of $\approx 8 \times 10^{-5}$ s$^{-1}$. Equating this value to $2V_{ex}/h$ (see above) we derive a nighttime deposition velocity for $H_2O_2$ of 0.8 cm s$^{-1}$. Given the assumptions made, the uncertainty related to this value is likely to be considerable. Nonetheless, the value obtained is

entirely consistent with nighttime deposition velocities of $0.8 \pm 0.2$, $1.0 \pm 0.3$ and $1.6 \pm 0.3$ cm s$^{-1}$ reported for $H_2O_2$ deposition over the Canadian boreal forest (Hall and Claiborn, 1997). Claiborn and Hall (1997) report daytime $H_2O_2$ deposition velocities that are a factor 10 ($\pm 5$) larger. Using this factor we calculate daytime deposition velocities of $8 \pm 4$ cm s$^{-1}$. This can be equated to $V_{ex}/h$, which assumes no gradient in $H_2O_2$ in a well-mixed boundary layer. Taking into account the





average daytime boundary layer height of 1400 m during HUMPPA, this converts to a loss rate constant of $(6 \pm 3) \times 10^{-5}$ s$^{-1}$ during the day. Later, we shall show that these values are consistent with results from the box-model based analysis of the dataset.

There is no data on PAA deposition, although Hall and Claiborn (Hall and Claiborn, 1997) measured deposition rates for

summed organic peroxides (mainly CH$_3$OOH) which were a factor of about two-to-three lower than for H$_2$O$_2$. We have thus adopted daytime loss rates of ROOH, CH$_3$OOH and PAA that are a factor two lower than for H$_2$O$_2$.  In the following, we present a more detailed discussion of the chemical factors which control the relative abundance of PAA and PAN during the campaign.

### 3.1 Production and loss of PAN and CH$_3$C(O)OOH

In the absence of a sufficiently rapid reaction with OH ($k < 3 \times 10^{-14}$ cm$^3$ molecule$^{-1}$ s$^{-1}$) and photo-dissociation (due to low cross sections and quantum yields in the tropospheric spectral range) (Talukdar et al., 1995), the lifetime of PAN during HUMPPA-COPEC-2010 was largely controlled by the temperature, which strongly affects its rate of thermal decomposition to CH$_3$C(O)O$_2$ and NO$_2$ (R2). The relative concentrations of trace gases such as NO, and peroxy radicals which can react with CH$_3$C(O)O$_2$ (via R2, R3, R4) to that of NO$_2$ which regenerates PAN (R1) and its dry deposition rate $d_{PAN}$ thus control

its effective lifetime. Unlike other peroxides (e.g. H$_2$O$_2$) PAA is not formed at significant yield in the ozonolysis of BVOCs such as α-pinene (Li et al., 2016).

The most important reactions (R1, R-1, R2a) describing the formation and loss of PAN and PAA involve the acetyl peroxy radical (CH$_3$C(O)O$_2$) including its reaction with NO (R3).

$$\text{CH}_3\text{C(O)O}_2 + \text{NO (+O}_2) \quad \rightarrow \text{CH}_3 + \text{CO}_2 + \text{NO}_2 \tag{R3}$$

PAA will be lost by reaction with OH (R4)

$$\text{OH} + \text{CH}_3\text{C(O)OOH} \quad \rightarrow \text{CH}_3\text{C(O)O}_2 + \text{H}_2\text{O} \tag{R4}$$

By analogy with reactions of OH with H$_2$O$_2$ and CH$_3$OOH, Orlando and Tyndall estimated $k_4$ to be ~ 1-5 $\times 10^{-12}$ cm$^3$ molecule$^{-1}$ s$^{-1}$ with CH$_3$C(O)O$_2$ (and H$_2$O) the predicted, dominant products following abstraction of the acidic H-atom in (R4)  (Orlando and Tyndall, 2003). Theoretical work (Rypkema and Francisco, 2013) suggests that H-abstraction at the

methyl group forming (in air) O$_2$CH$_2$C(O)OOH  is also possible. Recent experimental work (Wu et al., 2017) suggests that $k_4$ may be as large as $1.1 \times 10^{-11}$ cm$^3$ molecule$^{-1}$ s$^{-1}$ and this value has been adopted by the IUPAC panel (IUPAC, 2018) albeit with large associated uncertainty related to experimental difficulties.

The boundary layer lifetime of PAA and PAN will also be partially determined by deposition, especially in forests, as discussed above. The first-order rate constants representing dry deposition are $d_{PAN}$ and $d_{PAA}$. Experimental work (Wu et al.,

2015) on the uptake of PAA to ambient aerosol indicates an uptake coefficient (γ) of ≈ 3 $\times 10^{-4}$. Using the simplified expression (1) for the heterogeneous loss rate constant ($k_{het}$) of a trace gas to an aerosol particle:

$$k_{het} = 0.25\gamma\bar{c}A \tag{1}$$



where $\bar{c} \approx 28000$ cm s$^{-1}$ is the mean molecular velocity of PAA at room temperature and $A = 2 \times 10^{-6}$ cm$^2$ cm$^{-3}$ is the aerosol surface area density results in an approximate loss rate constant of $\approx 5 \times 10^{-6}$ s$^{-1}$ or a lifetime of about 3 days. Although PAA may be lost to aerosol in regions with extreme aerosol loading (Li et al., 2016) this loss term is negligible compared to e.g. depositional losses and reaction with OH in the boreal forest and is not considered further in this paper.

Considering only their in-situ, production and loss (i.e. ignoring advection) the net, relative rate of production of PAA and PAN (P$_{PAA}$ and P$_{PAN}$, respectively) is given by:

$$\frac{P_{PAA}}{P_{PAN}} = \frac{k_{2a}[CH_3C(O)O_2]_{ss}[HO_2] - [PAA](k_4[OH] + d_{PAA})}{k_1[CH_3C(O)O_2]_{ss}[NO_2] - [PAN](k_{-1} + d_{PAN})} \qquad (2)$$

where [CH$_3$C(O)O$_2$]$_{ss}$, is the steady state concentration of the acetylperoxy radical, determined by the ratio of its production and loss terms. On the right hand side of equation (2), the first terms of the numerator and denominator are readily recognised as governing PAA and PAN production, whilst the second terms are the summed loss rates. If both PAA and PAN were to acquire steady state, their relative concentrations would be given by:

$$\frac{[PAA]}{[PAN]} = \frac{[HO_2]}{[NO_2]} \cdot \frac{k_{2a}(k_{-1} + d_{PAN})}{k_1(k_4[OH] + d_{PAA})} \qquad (3)$$

Indicating that the PAA-to-PAN ratio depends on the relative abundance of HO$_2$ and NO$_2$. This is the expression we previously presented (Phillips et al., 2013) and which we used to derive very rough diel and campaign average HO$_2$ mixing ratios of about 30 ppt.

$$[HO_2] = \frac{[PAA]}{[PAN]} \cdot [NO_2] \cdot \frac{k_1(k_4[OH] + d_{PAA})}{k_{2a}(k_{-1} + d_{PAN})} \qquad (4)$$

Whether PAN and PAA acquire steady state (equivalence of production and loss terms in eqn. 2) depends on their loss rates. The first-order rate constant for PAN loss is given by the thermal decomposition rate constant ($k_{-1}$) multiplied by the fraction of the CH$_3$C(O)O$_2$ radical which does not reform PAN, plus the term for dry deposition, $d_{PAN}$, i.e.

$$\frac{-d[PAN]/dt}{[PAN]} = k_{-1} \cdot \frac{k_3[NO] + k_2[HO_2]}{k_3[NO] + k_2[HO_2] + k_1[NO_2]} + d_{PAN} \qquad (5)$$

At a temperature of 298 K and at 1bar pressure, $k_1 = 8.7 \times 10^{-12}$ cm$^3$ molecule$^{-1}$ s$^{-1}$, $k_{-1} = 3.8 \times 10^{-4}$ s$^{-1}$ and $k_2 = k_3 \approx 2 \times 10^{-11}$ cm$^3$ molecule$^{-1}$ s$^{-1}$. Using campaign average midday values of [NO$_2$] = 300 pptv, [NO] = 60 pptv and assuming [HO$_2$] = 20 pptv, we derive an effective PAN loss rate constant of $7.5 \times 10^{-5}$ s$^{-1}$ or a lifetime of $\approx 4$ hours without considering deposition. The PAN lifetime will however vary greatly over the diel cycle and will increase substantially during colder, dark periods as the thermal dissociation slows down. In the absence of HO$_2$ and NO the dominant fate of the CH$_3$C(O)O$_2$ radical at night is to react with NO$_2$ to reform PAN. The lifetime of PAN through the campaign was calculated using the measured NO and NO$_2$ concentrations and temperature and a model result for the HO$_2$ concentration (see section on box model below). Time dependent values of $d_{PAN}$ were derived using a sinusoidal variation of the boundary layer height during HUMPPA between ~1400 m during the day and ~200 m at night (Ouwersloot et al., 2012) which was matched in phase to the average diel temperature variation during the campaign. Deposition of PAN was calculated with loss rate constants of $2.3 \times 10^{-5}$ s$^{-1}$ at




night varying sinusiodally to a maximum value of $1.3 \times 10^{-5}$ at 15:30 (see above). The lifetime for PAN throughout the campaign is displayed in Figure 4.

Similarly, the loss rate of PAA will be controlled by its reaction with OH multiplied by the fraction that does not reform PAA (1-α), and its rate constant for dry deposition, $d_{PAA}$:

$$\frac{-d[PAA]/dt}{[PAA]} = k_4[OH] \cdot \frac{k_3[NO]+k_1[NO_2]+k_2(1-\alpha)[HO_2]}{k_3[NO]+k_1[NO_2]+k_2[HO_2]} + d_{PAA} \qquad (6)$$

The second term on the right hand side of this expression varies between about 0.9 during the day and 1.0 at night when, to a first approximation, NO and $HO_2$ are very low. With the same conditions as given above, using a model result for the diel variation of the OH concentration and taking $k_4$ to be $1 \times 10^{-11}$ cm$^3$ molecule$^{-1}$ s$^{-1}$ we derive an OH-induced PAA loss rate constant of $\approx 1 \times 10^{-5}$ s$^{-1}$, or a lifetime greater than 10 hours. Thus, both dry deposition and reaction with OH determine the

lifetime of PAA, though the dry deposition term dominates. The differences in the diel cycles of the PAA and PAN lifetimes are illustrated in Figure 4. Use of a single, average boundary layer height and the same variation in exchange velocities for each day during the campaign means that the lifetime of PAA shows a very uniform behaviour, varying between $\approx 5$ and 8 hours. In contrast, noon-time PAN lifetimes are sometimes less than three hours, but increase up to 16 hours at night.

Bearing in mind that the long lifetimes of PAA and PAN may partially invalidate the assumption of steady state, we can

insert noon-time values of PAA, PAN and $NO_2$ into expression (4) to derive $HO_2$ concentrations for each day during the campaign when data were available. The result is displayed in Figure 5, which also compares the $HO_2$ concentrations thus derived with direct measurements of LIF-$HO_2$ (Hens et al., 2014), available for only a limited period of the campaign.

The steady state calculations result in noon-time $HO_2$ concentrations between $5 \times 10^8$ and $2.5 \times 10^9$ molecule cm$^{-3}$. On most days on which PAA, PAN and LIF-$HO_2$ data are available (209, 210, 214, 218), the agreement is reasonable, exceptions

being days 211 and 217. Given that the PAA/PAN derived $HO_2$ concentrations are based on a steady state assumption and are heavily dependent on e.g. the PAA deposition rate, and the LIF-$HO_2$ data is likely to be an overestimate of $HO_2$ concentrations if high levels of organic peroxy radicals are present (Hens et al., 2014) the agreement may be coincidental.

A further cause for the occasional disagreement, also related to the long lifetimes of PAN and PAA, may be the advection of $NO_X$ from source that are too nearby for the PAN-to-PAA ratio to adapt and which will then result in an overestimation of

$HO_2$ (see equation 4). Below, we compare the $HO_2$ concentrations derived from equation (4) with those from a photochemical box model and show that good agreement is found during the warmer periods of the campaign (when daytime PAN lifetimes are shorter) but the disagreement can be significant during the colder periods when PAN is longer lived.

The $HO_2$ mixing ratio derived via equation (4) is rather insensitive to the rate coefficient for the reaction of OH with PAA, which is poorly constrained (IUPAC, 2018) or the concentration of OH if they lie within the expected ranges ($k_4 = 2\text{-}10 \times 10^{-}$

$^{12}$ cm$^3$ molecule$^{-1}$ s$^{-1}$, and [OH] = 5-20 $\times 10^5$ molecule cm$^{-3}$) reflecting the dominance of dry deposition losses of PAA such that $d_{PAA} > k_6[OH]$. Likewise, at noon-time, the thermal decomposition rate of PAN is too rapid for the deposition term to impact on the calculation of $HO_2$ ($k_{-1} \gg d_{PAN}$). The value of [$HO_2$] derived is however sensitive to the deposition rate of PAA chosen. An increase in $d_{PAA}$ by a factor of two leads to a similar increase in [$HO_2$].





A more detailed insight into the chemical and meteorological factors controlling PAA and PAN can be gained from use of a time-dependent, photochemical box-model constrained by some of the longer lived trace gases that contribute centrally to local photochemical generation of radicals and thus PAA and PAN. This is described below.

## 3.2 Box Model Description

The box-model was developed with the goal of simulating the concentration of PAA and $H_2O_2$ over several diel cycles, which requires realistic diel variation in the photochemical production and loss rates of the $CH_3C(O)O_2$ and $HO_2$ radicals. A highly simplified approach was taken in which the diel variation of $HO_2$ and $CH_3C(O)O_2$ radical concentrations over periods of several days was constrained by measured (non-radical) trace gases and photolysis constants, but overall radical levels were adjusted to optimise the simulation of PAA and $H_2O_2$. This may be seen as complementary to the modelling exercise

of Hens et al. (2014), who used a highly detailed chemical scheme and focussed on radical production in a bottom-up approach using a detailed reaction mechanism in which several biogenic organic trace gases were constrained.

In all model runs, the parameters directly constrained by observations were the temperature, concentrations of $O_3$, NO, $NO_2$, HONO, PAN, CO, HCHO, $CH_3CHO$, Σterpenes and the photolysis rate constants, $J$-O($^1$D), J-$NO_2$, J-HONO, J-$H_2O_2$ and J-HCHO. The model $CH_4$ mixing ratio was held constant at 1.8 ppmv. In sensitivity runs, the concentrations of trace gases that

were not measured (e.g. di-carbonyls, see later) were added to the model and their concentrations were calculated relative to those of related trace gases for which correlation is expected.

The complete reaction scheme is listed in Table S1 of the supplementary information. Rate coefficients were taken from the IUPAC evaluations (IUPAC, 2018). For the box-model, programmed in FACSIMILE code (Curtis and Sweetenham, 1987), several different scenarios were investigated, in which sections of the chemistry listed in Table S1 were deactivated or

modified in order to investigate sensitivity of the model output (i.e. concentrations of PAA, $H_2O_2$, $HO_2$ and OH) to certain reactions and assumptions made, and to optimise the simulation of the measured concentrations of these trace gases and radicals. The box model simulated the field data at 10 min resolution. Due to the constraint (by measurements) of relatively long-lived trace gases such as $O_3$ and $NO_X$, the model acquired steady state in less than one day. The simulation was initiated at day 196, the output only used from day 202 to day 220 (for which PAA and PAN data were available). Essential features

of the different model runs are listed in the section "box-model development" of the Supplementary information.

### 3.2.1 Model production and loss of $HO_X$ and ROx

OH was generated directly via $O_3$ photolysis, whereby the photolysis rate constant, $J$-O($^1$D), was modified to take into account the relative rates of quenching of the O($^1$D) atom by $N_2$ and $O_2$ and reaction with $H_2O$, i.e. it takes changes in humidity throughout the campaign into account. OH was also generated by photolysis of $H_2O_2$ and HONO and via reaction

of $HO_2$ with NO and $O_3$.

Loss of OH via reactions with the trace gases constrained by measurement such as CO, $H_2O_2$, HCHO, $O_3$ and $NO_2$ represent only a small fraction of the overall OH reactivity, which is dominated by organic trace gases (Nölscher et al., 2012). Thus,



two reactions in which OH was converted to organic peroxy radicals, were used to tune the model OH-reactivity and organic radical production rates. These OH-loss processes were parameterised as generic reactions of OH with biogenic trace gases and reactions with oxygenated volatile organic compounds (OVOCs) to form $RO_2^*$. In this case $RO_2^*$ is distinct from the sum of organic peroxy radicals which is usually denoted $RO_2$. The mixing ratios of the biogenic trace gases (terpenoids) was

constrained by PTR-MS measurements at high time resolution. The PTR-MS mixing ratios agreed with the summed terpenoids from the GC measurements.

A generic rate constant of $7 \times 10^{-11}$ $cm^3$ molecule$^{-1}$ s$^{-1}$ was used for the reaction of OH with terpenes. This was calculated from the relative concentrations of the terpenes as measured by GC and weighted with their individual rate constants for reaction with OH (Atkinson and Arey, 2003b; IUPAC, 2018).

The mixing ratios of OVOCs were linked by a factor (adjusted to obtain good agreement between observation and model output) to measurements of $CH_3CHO$ and HCHO: For the reaction of OVOC with OH we used a rate coefficient of $1 \times 10^{-11}$ $cm^3$ molecule$^{-1}$ s$^{-1}$. This approach is different to that taken by Hens et al. (2014) who tied missing OH-reactivity to an unknown trace gas that behaved like $\alpha$-pinene but at concentrations that were a factor 5-10 higher than measured. Tying missing OH-reactivity to $\alpha$-pinene could however not simulate the high OH-reactivities during the biomass burning

impacted periods of the campaign.

The modelled OH-reactivity varied between 3 and 50 s$^{-1}$, the highest values being associated with the biomass-burning impacted days as observed. A comparison between modelled and measured OH-reactivity reveals broad agreement (Figure 6) with several high reactivity events captured by the model. The main exceptions are the events on day 202.5 (OH-reactivity up to 43 s$^{-1}$) and day 215 (peak OH-reactivity of 70 s$^{-1}$). Figure 6 also plots the time series of the organic content of aerosol.

This varies in a similar manner to the modelled OH reactivity. This is expected as increased OH reactivity should lead to larger rates of generation of condensable OVOCs and thus SOA mass. On the other hand, it will also reflect that the high values of OH reactivity are tied in the model to high levels of measured OVOCS ($CH_3CHO$ and HCHO), which may also be correlated with organic aerosol content. Generally, the variations in the measured OH reactivity are paralleled by changes in the organic aerosol content. The most prominent exceptions are again the very high OH reactivities at day 202.6 (~43 s$^{-1}$)

and day 214.6 (~70 s$^{-1}$). No corroborative evidence for high reactivity on these days could be found in the extensive datasets available for this campaign. For example, there were no significant increases in any measured trace gases (biogenic or biomass burning related), no reductions in the OH levels or enhanced formation of products such as HCHO. Instrument failure and subsequent tests resulted in an interruption in OH-reactivity measurements shortly before the sharp increase in OH-reactivity on day 214 was measured. Though it is not obvious why this should have resulted in such a large, positive bias

when measurements were resumed, the mismatch with other data, proximity to the power-down and exponential decay in the reactivity suggest that the instrument was not fully operational in this period.

$RO_2^*$ reacted in the model with NO to form $HO_2$ (generic rate coefficient of $1 \times 10^{-11}$ $cm^3$ molecule$^{-1}$ s$^{-1}$) and with $HO_2$ to form peroxides (ROOH, with generic rate coefficient of $0.8 \times 10^{-11}$ $cm^3$ molecule$^{-1}$ s$^{-1}$) and regenerate OH at $0.2 \times 10^{-11}$ $cm^3$





molecule$^{-1}$ s$^{-1}$. As the chemical identity of the OVOCS are unknown a more explicit description of the chemistry was not warranted.

HO$_2$ was generated directly via photolysis of HCHO and from reactions of OH with O$_3$, CO, HCHO and H$_2$O$_2$, and the reaction of NO with CH$_3$O$_2$ and other peroxy radicals. The most important HO$_2$ reactions in the model are its self-reaction to

form H$_2$O$_2$, and reaction with NO and O$_3$ to form OH. It also reacts with the CH$_3$C(O)O$_2$ radical (forming PAA and acetic acid and recycling OH). The uptake of HO$_2$ to aerosol has been reported to be important for the HO$_X$ budget under some conditions (Jacob, 2000; Liang et al., 2013). The present IUPAC evaluation (Ammann et al., 2013; IUPAC, 2018) for the heterogeneous interaction of HO$_2$ with aqueous aerosol has a dependence on the HO$_2$ concentration as well as the aerosol pH which is based on the formulation of Thornton et al. (2008). For average HUMPPA-COPEC conditions a value for the

uptake coefficient ($\gamma$) between $10^{-3}$ and $10^{-4}$ is estimated. In combination with equation (2), and for HO$_2$ = 5 × $10^{8}$ molecule cm$^{-3}$ ($\approx$ 20 pptv) this results in a loss rate for HO$_2$ of $\approx$ 1 × $10^{4}$ – 1 × $10^{5}$ molecule cm$^{-3}$ s$^{-1}$, which is insignificant compared to HO$_2$ loss rates via its self-reaction (1.5 × $10^{6}$ molecule cm$^{-3}$ s$^{-1}$) and reaction with NO and (1.2 × $10^{7}$ molecule cm$^{-3}$ s$^{-1}$ with NO = 100 pptv) close to midday. Heterogeneous losses of HO$_2$ were therefore not included in the model.

The rate of CH$_3$C(O)O$_2$ formation in the model is determined by the rate of oxidation or photolysis of its organic precursors,

including the oxidation of CH$_3$CHO via reaction with OH:

$$OH + CH_3CHO + O_2 \rightarrow CH_3C(O)O_2 + H_2O \tag{R5}$$

As we describe later, this reaction alone was not sufficient to describe the formation of PAA at the levels observed and the model was augmented with other CH$_3$C(O)O$_2$ sources such as the degradation of photo-labile carbonyl species. Tying CH$_3$C(O)O$_2$ production rates to the levels of oxidised VOCs as measured by the PTRMS was found to be especially

important during biomass burning influenced periods in which VOCs (e.g. propane, butane) and OVOCs (e.g. CH$_3$CHO, CH$_3$C(O)OH, CH$_3$OH) were also greatly enhanced.

Finally, in order to test potential radical sources leading to enhanced HO$_X$ concentrations at night-time, formation and reactions of the NO$_3$ radical were implemented in a simplified manner (See Table S1) as was the formation of OH via ozonolysis of alkenes.

**3.2.2 Model production of H$_2$O$_2$ and PAA**

PAN, PAA and H$_2$O$_2$ are NO$_X$ dependent, stable products of HO$_X$, ROx and NO$_X$ interactions. The goal of the modelling study is to examine whether their measurement can provide insight into the HO$_X$ budget in the boreal environment by adapting a reaction scheme to match the observed mixing ratio of H$_2$O$_2$ and PAA over chemically different periods of the campaign. H$_2$O$_2$ was generated in the model via the HO$_2$ self-reaction, the rate constant of which displays a complex

dependence on temperature, pressure and humidity (IUPAC, 2018). The rate coefficient during the campaign varied between 4.3 and 7.2 × $10^{-12}$ cm$^3$ molecule$^{-1}$ s$^{-1}$. We also include H$_2$O$_2$ formation from the ozonolysis of terpenes.



Model PAA was generated only in the reaction between $HO_2$ and $CH_3C(O)O_2$ radicals as described above. PAA and $H_2O_2$ may be lost via photolysis, reaction with OH, dry deposition and uptake to aerosol. As described in section (3.1) for PAA the first-order loss rate constant for heterogeneous reaction of $H_2O_2$ with aerosol particles can be calculated from expression (2) using $\bar{c} \approx 44000$ cm s$^{-1}$, $A = 2 \times 10^{-6}$ cm$^2$ cm$^{-3}$ and $\gamma = 3 \times 10^{-4}$ (Wu et al., 2015), resulting in $k_{het} = 7 \times 10^{-6}$ s$^{-1}$. As concluded for PAA, loss of gas-phase $H_2O_2$ via uptake to aerosol is insignificant compared to dry deposition.

### 3.2.3 Dry Deposition

Depositional loss of PAA and $H_2O_2$ was simulated in the model using time dependent first order loss rate constants that were derived assuming a sinusoidal variation of the boundary layer height during HUMPPA between ~1400 m during the day and ~200 m at night (Ouwersloot et al., 2012) which was matched in phase to the average, diel temperature during the campaign. Deposition rate constants were calculated from deposition velocities from the literature scaled to the observed loss rate of PAA and $H_2O_2$ at night (see section 3). For $H_2O_2$, the data of Hall and Claiborn (Hall and Claiborn, 1997) were used. Their values are chosen as they are expected to be most relevant for the Boreal tree type and forested environment but are still in broad agreement with other datasets for $H_2O_2$ deposition (Stickler et al., 2007). The daytime deposition of PAA was modelled to be 50 % that of $H_2O_2$, with the same variation over the diel cycle. The diel cycle of the deposition velocities for PAA and $H_2O_2$ are displayed in Figure S2 of the Supplementary Information.

### 3.3 Box Model Results

The constrained model, with the photochemical sources of radicals described in section 3.2.1 significantly underestimated the observations of $H_2O_2$ and PAA, especially during the biomass burning episodes. Addition of an extra source of $CH_3C(O)O_2$ in the reaction of OH with OVOC (at 30% branching ratio) was found not to help as even small increases (less than factor of two) in the modelled concentration of PAA could only be achieved at the cost of reducing the OH, $HO_2$ and thus $H_2O_2$ concentrations significantly below the measurements. This is readily understood as PAA is an effective sink of $HO_X$ for this low $NO_X$ environment.

The PAA production rate was therefore enhanced by introducing a $CH_3C(O)O_2$ production term that does not require initiation by reaction with OH, i.e. photolysis of a precursor trace gas, which results in $CH_3CO$ release. Acetone ($CH_3C(O)CH_3$) was present at mixing ratios of up to 10 ppbv during the periods impacted by biomass burning, yet its slow photolysis (lifetime of months) means that it is not a significant source of $CH_3C(O)O_2$ in the boundary layer. In contrast, dicarbonyls such as methyl-glyoxal ($CH_3C(O)CHO$), pyruvic acid ($CH_3C(O)C(O)OH$) or biacetyl ($CH_3C(O)C(O)CH_3$) have absorption spectra that extend beyond 400 nm and are rapidly photolysed (lifetime at noon of a few hours for $CH_3C(O)CHO$ and $CH_3C(O)C(O)OH$ and 15 mins for $CH_3C(O)C(O)CH_3$) and may represent an efficient source of $CH_3C(O)O_2$ and $HO_2$ if present at sufficient concentrations.

$$CH_3C(O)CHO + h\nu\ (O_2) \qquad \rightarrow CH_3C(O)O_2 + HO_2 + CO \qquad\qquad (R6)$$





$$CH_3C(O)C(O)CH_3 + h\nu \ (O_2) \quad\quad \rightarrow 2 \ CH_3C(O)O_2 \quad\quad\quad\quad\quad\quad\quad\quad\quad (R7)$$

$$CH_3C(O)C(O)OH + h\nu \ (O_2) \quad\quad \rightarrow CH_3C(O)O_2 + HO_2 + CO_2 \quad\quad\quad\quad\quad (R8)$$

Addition of short-lived hydrocarbons including methyl-glyoxal has been found to increase $CH_3C(O)O_2$ radical production rates in models of biomass burning plumes (Fischer et al., 2013) and to dramatically increase PAN production rates in a global model (Ito et al., 2007).

Methyl-glyoxal is formed at high yield from the OH-initiated oxidation of several biogenic (especially isoprene) and anthropogenic VOCs e.g. via degradation of aromatic hydrocarbons (Arey et al., 2009; Obermeyer et al., 2009) and is found in biomass burning impacted air masses (Fu et al., 2008; Akagi et al., 2011; Stockwell et al., 2015) and those influenced by urban emissions of aromatics (Liu et al., 2010). It is also formed at yields of a several percent from the ozonolysis of mono-terpenes such as α-pinene and $\Delta^3$-carene (Yu et al., 1998; Fick et al., 2003), which were both present at high concentrations during HUMPPA-COPEC-2010. Model estimates of PAN formation suggest that, on a global scale, 30 % of the acetyl peroxy radical generation is the result of methyl-glyoxal degradation, with 44 % arising via $CH_3CHO$ oxidation by OH (Fischer et al., 2013).

Biacetyl is formed in the photo-oxidation of aromatic hydrocarbons in the presence of $NO_X$ (Atkinson et al., 1980; Arey et al., 2009; Obermeyer et al., 2009) and is also expected to be present in biomass-burning impacted air-masses. The proposed intermediacy of dicarbonyls such as methyl-glyoxal and biacetyl in forming PAA in this study is consistent with the dominant role for di- and tri-alkyl benzenes in PAA formation found by Zhang et al (Zhang et al., 2010). Pyruvic acid is formed in the gas-phase degradation of biogenic and anthropogenic hydrocarbons and has been observed at mixing ratios of hundreds of pptv (Andreae et al., 1987; Jacob and Wofsy, 1988; Talbot et al., 1995; Veres et al., 2011). As a precursor in the biosynthesis of terpenoids, pyruvic acid can be directly emitted by vegetation resulting in very large mixing ratios (several ppbv) in a non-oxidative environment (Jardine et al., 2010). It is also found in secondary organic aerosol and may be formed in the aqueous phase reaction of methyl-glyoxal (Tan et al., 2012). In a recent campaign (IBAIRN, 2016) at the Hyytiälä site, our Iodide-CIMS instrument monitored large signals at $m/z = 87$, which we assigned to the $CH_3C(O)CO_2^-$ ion from pyruvic acid. Post-campaign calibration of the CIMS resulted in mixing ratios of several hundred pptv, which were correlated with other biogenic trace gases. We thus have direct evidence of pyruvic acid at this site, albeit in the autumn (when emissions are likely to be weaker) rather than in the summer.

A further, short lived di-carbonyl that is formed in the OH and $O_3$-initiated degradation of several VOCs including alkenes, acetylene and aromatics (Calvert et al., 2000; Volkamer et al., 2001; Calvert et al., 2002) and which is present in biomass burning (Fu et al., 2008) is glyoxal (HC(O)C(O)H). It is also formed at high yield (via glycol aldehyde) in the OH oxidation of methyl butenol (Atkinson and Arey, 2003a), emitted by pine trees and therefore of relevance for the HUMPPA-COPEC campaign. The photolysis of glyoxal is efficient (lifetimes at noon of ~ 4 hrs) and results in formation of two $HO_2$ radicals:

$$HC(O)CHO + h\nu \ (O_2) \quad\quad \rightarrow 2 \ HO_2 + 2 \ CO \quad\quad\quad\quad\quad\quad\quad\quad\quad\quad (R9)$$



In the absence of direct measurements, indications of the presence of biacetyl and methyl-glyoxal were sought in the PTR-MS dataset. Strong increases in mixing ratio at $m/z = 73$ and $m/z = 87$ were observed which would correspond to protonated methyl-glyoxal and biacetyl, respectively. Indeed, most PTR-MS masses monitored displayed similar trends, with strong increases during the biomass burning impacted periods. Campaign time series for selected masses are shown in the

supplementary information (Figure S1).

For $m/z = 73$ a peak mixing ratio of $\approx 0.8$ ppbv was observed during the first biomass burning event (day 206-212). At the same time a peak value of 0.5 ppbv was reported for mass 87. The PTRMS signal at mass 73 is usually considered to be methyl ethyl ketone (de Gouw and Warneke, 2007; Blake et al., 2009; Yanez-Serrano et al., 2016), a product of butane degradation. n-butane was enhanced during the biomass burning periods and correlated with $m/z = 73$, so that a significant

(or dominant) contribution of methyl ethyl ketone to this mass is probable.

The PTRMS mass 87 (peaking at mixing ratios of 0.5 ppb during the biomass burning impacted periods) is usually assigned to 2-methyl-3-buten-2-ol (MBO) which is emitted from coniferous trees (Schade et al., 2000) but may have contribution from several trace gases including C5-carbonyls, butadione and methacrylic acid (de Gouw and Warneke, 2007; Blake et al., 2009; Muller et al., 2016). We are not aware that biacetyl has been reported at this mass.

Previous measurements of methyl-glyoxal and glyoxal in rural locations have revealed mixing ratios up to a few hundred pptv (see Fu, 2008 for a summary) which were correlated with those of HCHO (Lee et al., 1995) or CO (Spaulding et al., 2003) and which had mixed biogenic and anthropogenic sources. Greatly elevated concentrations of methyl-glyoxal (>900 pptv) and glyoxal (> 500 pptv.) have been observed in biomass burning impacted air masses in a rural environment (Kawamura et al., 2013). Even at sub 100 pptv levels, methyl-glyoxal was found to contribute significantly to $CH_3C(O)O_2$

radical production (Lee et al., 1995). We note that the PTR-MS is insensitive to glyoxal due to instability of the protonated parent-ion at $m/z = 59$ (Stoenner et al., 2017) which is dominated by acetone.

The potential impact of dicarbonyls on the $HO_X$ and $CH_3C(O)O_2$ budget was investigated by incorporating methyl-glyoxal, biacetyl, glyoxal and pyruvic acid into the model with the concentrations of the first three di-carbonyls constrained by a correlation factor with $CH_3CHO$. Mixing ratios of pyruvic acid at this site (in autumn) have been found to be correlated with

(and at similar concentration to) those of directly emitted biogenics and its concentration was arbitrarily set equal to that of the summed terpenes.

The ratios of $CH_3CHO$ to methyl-glyoxal, biacetyl and glyoxal were varied to optimise the simulation of the measured PAA and $H_2O_2$ mixing ratios and resulted in maximum concentrations of methyl-glyoxal, glyoxal and biacetyl in the model of approximately 0.75, 0.15 and 0.22 ppbv, respectively, which were present during the peak of the biomass burning impacted

episodes (when $CH_3CHO$ levels were largest). The relative mixing ratios of di-carbonyls is not important for the model result, i.e. a reduction in the methyl-glyoxal mixing ratio would result in a decrease in both $CH_3C(O)O_2$ and $HO_2$ radical production, which could be balanced by an increase in pyruvic acid (with also generates one of each radical) or by an increase in both biacetyl and glyoxal, scaled by the relative J-values. Based on experimental absorption spectra and quantum yields, the photolysis rates of pyruvic acid, glyoxal, methyl-glyoxal and biacetyl were modelled as factors of 0.033, 0.0076,



0.019 and 0.208 that of $NO_2$, which absorbs light in a similar region of the spectrum. We note that large differences in experimental results (see (IUPAC, 2018) for a summary) and in the preferred values of evaluation panels (Burkholder et al., 2015; IUPAC, 2018) indicate that the J-values of the di-carbonyls are associated with significant uncertainties, possibly as large as a factor two.

The incorporation of dicarbonyls into the model has the anticipated effect of increasing acetyl and $HO_2$ peroxy production rates and thus levels of PAA, especially during the biomass burning impacted periods. As seen in Figure 7, the model does a reasonable job of reproducing both PAA and $H_2O_2$. Especially encouraging is the good model-measurement agreement in the period between day 204 and 209, in which PAA levels increased by an order of magnitude and which were accompanied by large increases in $H_2O_2$ and the generally good agreement with OH measurements. The measurement-model agreement is

further exemplified in Figure 8 which plots the diel profiles of PAA and $H_2O_2$ and OH and the comparison with the model output for the same time period. Considering that we use a diel cycle for the deposition term that is unchanged for the whole campaign (i.e. does not vary with relative humidity, wind-speed, rates of turbulent mixing etc.) and thus does not reflect any variation in the meteorological situation at the site, the agreement for $H_2O_2$ and PAA is very good and, within the observed variability and considering the experimental uncertainty in the measurements, the model captures the average variation of

each trace gas across the diel cycle. It is also noteworthy, that the model captures the non-zero OH levels at nighttime as well as the noon-time maximum values. This contrasts with the conclusions of Hens et al. (2014), who also modelled the reaction of ozone with biogenics (but using speciated terpenes from GC-measurements) and were unable to generate and sustain sufficient OH at nighttime to match the observations. Although the difference may be partially related to our use of PTRMS measured terpenes and an average yield of OH based on latest recommendations for individual terpenes, the main effect is

the nighttime recycling of $HO_2$ and OH via reactions of peroxy radicals formed from OH generated in ozonolysis reactions. Given the high OH-reactivities measured during the HUMPPA-COPEC-2010 campaign, the fate of most OH radicals formed at nighttime will be reaction with terpenes and other VOCs to form $RO_2^*$. At nighttime, in the absence of NO, the main fates of peroxy radicals are reaction with $HO_2$ and self-reaction. In the model, the reaction of $RO_2^*$ with $HO_2$ forms ROOH (80 %) with a minor (20 %) reaction pathway forming OH as observed in laboratory studies (at yields of 10-60 %) for several

substituted oxidised organic peroxy radicals (Dillon and Crowley, 2008; Jenkin et al., 2010; Groß et al., 2014). The self-reaction of $RO_2^*$ is modelled to form $HO_2$ (via the degradation of the alkoxy radicals formed in the first step) so that non-zero concentrations of, $HO_2$ and $RO_2^*$ are predicted to be present at night. The modelled, $HO_2$ and $RO_2^*$ concentrations are displayed in Figure 9a. The initially surprising result is that $RO_2^*$ radicals are present at night at concentrations that are comparable to, or even exceed those predicted for the daytime. This reflects the weak sinks of $RO_2^*$

at night time (owing to the absence of NO), rather than high rates of production. The diel cycles of $RO_2^*$ (Figure 9d) and $HO_2$ (Figure 9c) are very different. Whereas the $HO_2$ concentration is defined by its photochemical production, with a maximum coinciding with the maximum of the actinic flux and low concentrations at night, that of $RO_2^*$ shows two distinct, broad maxima, one at noon and one at midnight, both at concentrations of $\approx 5 \times 10^8$ molecule $cm^{-3}$ ($\approx 20$ pptv). The modelled





$RO_2$*-to-$HO_2$ ratio is ≈ 1 at 12:00 UTC, increasing to ≈ 5 at midnight. This is consistent with the box-model result of Hens et al. (2014) who report $RO_2$-to-$HO_2$ ratios of between 0.5 and 4.5.

$RO_2$* production at night occurs via OH (formed via ozonolysis of alkenes and reaction of $O_3$ with $HO_2$) with OVOC and terpenes, the weak sinks enabling it to build up in concentration. The modelled, rapid depletion of $RO_2$* at sunrise is a result

of increasing NO concentrations, which is followed by a second maximum in the $RO_2$* concentration at noon resulting from photochemical generation of OH. Organic radicals have previously been observed at nighttime in several locations and are often attributed to the $O_3$ initiated oxidation of organics (Hu and Stedman, 1995; Reiner et al., 1997a; Salisbury et al., 2001; Geyer et al., 2003; Emmerson and Carslaw, 2009; Sommariva et al., 2011; Andrés-Hernández et al., 2013). Nighttime concentrations of RO$x$ ($HO_2$ + $RO_2$ in molecule cm$^{-3}$) of 5-7 × 10$^8$ (Reiner et al., 1997a), 1-2 × 10$^9$ (Andrés-Hernández et al.,

2013), ≈ 2 × 10$^8$ (Geyer et al., 2003) and ≈ 4 × 10$^8$ (Hu and Stedman, 1995) have been reported, which are consistent with the model results discussed here, albeit for chemically different locations. The double maximum in $RO_2$* is also consistent with the data of Geyer et al. (2003) who observed a similar double maximum in RO$x$ at noon and late evening.

Figure 7 indicates that the measured, midday values of LIF-$HO_2$ are generally larger than modelled $HO_2$ and that the diel variation in LIF-$HO_2$ (including large nighttime values) is incompatible with the modelled $HO_2$ profile (red line). LIF-$HO_2$

measurements as measured during HUMPPA-COPEC 2010 are prone to interference from organic peroxy radicals (Fuchs et al., 2011; Fuchs et al., 2016; Lew et al., 2018) and the predicted presence of large concentrations of $RO_2$* at nighttime has important consequences for our interpretation of the LIF-$HO_2$ data set. In Figure 9b, we plot the LIF-$HO_2$ concentrations along with modelled $HO_2$ with a fractional contribution from $RO_2$* (i.e. [$HO_2$]$_{model}$ + 0.35[$RO_2$]$_{model}$).

The result (red line) reproduces not only the high apparent nighttime values of $HO_2$ observed but also the non-linear

dependence on actinic flux and the higher values during the episodes impacted by biomass burning. Assuming that the $RO_2$ are detected as $HO_2$ with an efficiency of 35 % thus reconciles measurement and model results, and is consistent with detection efficiencies for longer chain organic peroxy radicals.

Figure 10 shows the modelled time-series of $RO_2$*, $CH_3O_2$ and $CH_3C(O)O_2$. The organic peroxy radicals formed from reaction of OH with OVOCs and terpenes ($RO_2$*) are most abundant, reflecting the fact that the OH reactivity in the model is

dominated by these trace gases. $CH_3O_2$ is also present at high concentrations and Figure S4 of the supplementary information plots the time dependent reactive flux through each of the reactions in the model that generate it. The dominant reactions are those of the $CH_3C(O)O_2$ radical, which reacts via formation of $CH_3C(O)O$, which decarboxylates to form the $CH_3$ radical and thus (via $O_2$ addition) $CH_3O_2$. In comparison, the formation of $CH_3O_2$ via $CH_4$ oxidation represents, on average, less than 10 % of the total.

Organic peroxides (ROOH) other than PAA are important indicators of coupling between organic peroxy radicals and $HO_2$. The smallest member of the ROOH family is $CH_3OOH$, formed in the reaction between $CH_3O_2$ and $HO_2$:

$CH_3O_2$ + $HO_2$                   → $CH_3OOH$ + $O_2$                                   (R10)

Larger ROOH are also formed via e.g. the OH initiated oxidation of VOCs (e.g. terpenes, ($C_{10}H_{16}$) in air:





$$OH + C_{10}H_{16} \; (+ O_2) \qquad \rightarrow HOC_{10}H_{16}O_2 \qquad\qquad (R11)$$

$$HOC_{10}H_{16}O_2 + HO_2 \qquad \rightarrow HOC_{10}H_{16}OOH + O_2 \qquad\qquad (R12)$$

$$HOC_{10}H_{16}O_2 \qquad (+ O_2) \qquad \rightarrow HOM + OH \qquad\qquad (R13)$$

The conventional formation of ROOH, from reaction of the terpene derived peroxy radical via reaction with $HO_2$, may be

augmented for some terpenes by auto-oxidation to also release OH and form a highly oxygenated molecule (HOM), that may

partition substantially to the particle phase (Cantrell et al., 1997; Reiner et al., 1997b).

Speciated organic peroxides were not measured during the campaign, but an indication of their levels was gained from the

measurement of total organic peroxides (see section 2), which is biased towards those which are soluble and will thus have a

significant contribution from PAA. In Figure 11 we plot (red line) the model predicted time series of summed PAA and

$CH_3OOH$ mixing ratios after taking into account their known collection efficiencies in the scrubber (~90 % for PAA, 60 %

for $CH_3OOH$). Model $CH_3OOH$ is formed only in reaction (R10) and that the sources of the $CH_3O_2$ precursor are limited to

the reactions (R2c) and (R3), which both initially generate $CH_3CO_2$ , which decomposes to $CH_3$ and thus $CH_3O_2$ in the

presence of $O_2$ as well as (R14) and (R15):

 and R14- R16) below:

$$OH + CH_4 \; (+ O_2) \qquad \rightarrow CH_3O_2 + H_2O \qquad\qquad (R14)$$

$$OH + CH_3OOH \qquad \rightarrow CH_3O_2 + H_2O \qquad\qquad (R15)$$

The removal of $CH_3OOH$ in the model is mainly due to reaction with OH and dry deposition, the latter being set to 50 % that

of $H_2O_2$. Figure 11 reveals that the transition from low to high mixing ratios of $CH_3OOH$ during the pre- and post-biomass

burning periods is adequately reproduced in the model and, given the assumptions made above, agreement in terms of

absolute concentrations is also satisfactory. The blue line in Figure 11 was obtained by assuming that model derived ROOH

is also detected but with a reduced efficiency of 0.1. The modelled sum of PAA and $CH_3OOH$ is close to the measured sum

of organic peroxides, indicating that these peroxides dominate the total organic peroxide measured by the instrument.

Detection of other peroxides, even with low sensitivity (blue line), results in a model over-prediction. This likely reflects the

fact that model generated ROOH are much less soluble than PAA or $CH_3OOH$ and thus only a small fraction is detected.

Alternatively, peroxides with large, substituted organic groups (e.g. those formed from terpenes) are likely to partition to the

aerosol phase.

Correlation diagrams (day 202 to 220) of measured versus modelled mixing ratios and concentrations of PAA, $H_2O_2$, ROOH,

$HO_2$ and OH are displayed in Figure 12. The agreement between modelled and measured PAA, OH and $H_2O_2$ is generally

good, with slopes that are, within the error limits of the measurements, equivalent to one. The exception is LIF-$HO_2$ (slope of

0.48) which is the result of an interference in the measurement as described above.

$RO_2$ radicals may also be formed at night in the $NO_3$ initiated oxidation of organics (Salisbury et al., 2001; Emmerson and

Carslaw, 2009; Sommariva et al., 2011; Andrés-Hernández et al., 2013). $NO_3$ is formed via the reaction between $NO_2$ and $O_3$



and in forested areas, in the absence of strong, local NO emissions, will be lost mainly via reaction with unsaturated hydrocarbons such as terpenes forming nitrated peroxy radicals (N-RO$_2$)

$$NO_2 + O_3 \qquad\qquad \rightarrow NO_3 + O_2 \qquad\qquad\qquad\qquad (R16)$$

$$NO_3 + VOC\ (+ O_2) \qquad\qquad \rightarrow N\text{-}RO_2 \qquad\qquad\qquad\qquad (R17)$$

Like other peroxy radicals these can react with NO or NO$_3$ or HO$_2$ eventually forming a variety of multi-functional, organic nitrates (Ng et al., 2016). The reaction with NO proceeds via an alkoxy radical, which, via reaction with O$_2$, may also form HO$_2$. As HO$_2$ reacts with NO$_3$ to form OH radicals, NO$_3$ reactions can represent a nighttime source of HO$_X$ radicals (Platt et al., 1990). However, due to the high concentrations of terpenes measured during night in HUMPPA-COPEC-2010 the steady state concentration of NO$_3$ was reduced to less than one pptv (Rinne et al., 2012) so that reaction with HO$_2$ or RO$_2$ will be

reduced in importance. As PAN and thus CH$_3$C(O)O$_2$ are present at night, the reaction between HO$_2$ and CH$_3$C(O)O$_2$ is also an efficient route to OH (R2c). In both scenarios, NO$_3$ and CH$_3$C(O)O$_2$ can be considered to perform the same job as NO during the day: conversion of HO$_2$ to OH. Model NO$_3$ mixing ratios were less than 0.1 pptv, consistent with upper limits reported in previous NO$_3$ measurements at this site (Rinne et al., 2012; Liebmann et al., 2018). In agreement with Hens et al. (2014) we find that NO$_3$ reactions are an insignificant source of nighttime HO$_X$ and RO$_2$ during HUMPPA-COPEC-2010.

Our photochemical box-model does not explicitly take into account the effect of early morning entrainment of trace gases such as H$_2$O$_2$ and PAA from the residual layer. We have examined the potential effect of entraining H$_2$O$_2$ by simulating its down-mixing from higher (above canopy) levels that are likely to be richer in H$_2$O$_2$ due to the absence of dry deposition in air-masses disconnected from the ground. Entrainment of H$_2$O$_2$ from the residual layer would increase its canopy level mixing ratio in the morning when ground heating results in mixing of the PBL and the overlying layers. The point at which

entrainment starts each morning was defined by the time at which the trend in temperature at 16.8 m (close to canopy top) switched from negative to positive. This was usually between about 05:00 and 06:00 UTC, but variable owing to variable cloud cover. The concentration of H$_2$O$_2$ in the upper layers was taken to be same as that modelled for the previous afternoon at 16:00 when insolation weakens and the residual layer starts forming. The effect of entrainment was coded in the model by allowing a proxy for H$_2$O$_2$ (at the H$_2$O$_2$ concentration from the previous afternoon) to decay to form H$_2$O$_2$ at a constant rate

until the temperature at canopy top stopped increasing (usually over a period of ≈ 6 hours) and the boundary layer was well mixed. The effect of the modelled entrainment was to skew the diel profile of H$_2$O$_2$ to larger pre-noon values and away from the average trend observed. While it is clear that entrainment from higher layers can increase pre noon mixing ratios, it appears that this effect is much weaker than the photochemical formation of H$_2$O$_2$ during this campaign.

### 3.3.1 Model Sensitivity to PAA losses

Due to lack of experimental data, there is some uncertainty associated with the physical constants describing the main loss processes for PAA. To our knowledge, neither absolute nor relative deposition velocities have been measured for PAA, and the preferred value of the rate constant for its reaction with OH carries an uncertainty of a factor of two (IUPAC, 2018). The nighttime loss-rate constants for H$_2$O$_2$ and PAA were derived from measurements but the daytime values were scaled using





literature data for $H_2O_2$ as described above. So far, we have assumed that the daytime deposition velocity of PAA is 50% that of $H_2O_2$. If deposition is not limited by transport but by surface resistance this factor may be smaller as PAA is less soluble than $H_2O_2$. In a sensitivity study, we therefore reduced the daytime deposition velocity of PAA to just 25% that of $H_2O_2$. The nighttime value was held at the measured value. As expected, the modelled PAA mixing ratios increased with the model-to-

measurement ratio also increasing from 0.86 to 1.09. Considering the total uncertainty of the PAA measurements, both results are however statistically indistinguishable from 1 and can be considered to be in satisfactory agreement.

The sensitivity of the model output to the OH-rate coefficient was tested by reducing it by a factor of two from the IUPAC recommended value of $1.1 \times 10^{-11}$ $cm^3$ $molecule^{-1}$ $s^{-1}$. This is the lower limit of the IUPAC recommendation and is close to previous estimates based on analogous reactions of OH with organic peroxides. As expected, a reduction in the loss term

increased the modelled PAA mixing ratio, the model-to-measurement ratio also increasing slightly from 0.86 to 0.92. The weak sensitivity to the OH + PAA rate coefficient in the range $5-11 \times 10^{-12}$ $cm^3$ $molecule^{-1}$ $s^{-1}$ is expected as this reaction contributes only a small fraction of the overall loss of PAA, which is dominated by dry deposition. On the other hand, increasing the OH-rate coefficient to $2.2 \times 10^{-11}$ $cm^3$ $molecule^{-1}$ $s^{-1}$ results in this reaction becoming competitive with dry deposition, with a reduction in the model-to-measurement ratio from 0.86 to 0.77.

**3.3.2 Modelled sources of radicals during HUMPPA-COPEC-2010**

Having constructed a photochemical chemical scheme that can 1) mimic the diel behaviour of several traces gases (PAA, $H_2O_2$ and, with some caveats, $CH_3OOH$) that are strongly linked to $HO_2$, and $RO_2$ concentrations, and 2) which can reproduce the measured concentrations of OH and $HO_2$, we can use the model to identify the relative contributions of the processes forming OH, $HO_2$ and $CH_3C(O)O_2$ during the campaign. The results are summarised as campaign averages (days

202-220) in Figure 13.

Fig. 13c indicates that the major source of the acetylperoxy radical in the model is the photolysis of the di-carbonyls biacetyl, methyl-glyoxal and pyruvic acid, together accounting for 61 %, with almost all of the remaining $CH_3C(O)O_2$ coming from PAN decomposition. The reaction of $CH_3CHO$ with OH accounted for only 4 %.

For $HO_2$ (Fig. 13b) the picture is more complex with direct (primary) formation via the photolysis of formaldehyde, methyl-

glyoxal and pyruvic acid accounting for 24 %, the major source being the reaction between organic peroxy radicals and NO (64 %). This observation is consistent with the conclusions of (Kim et al., 2013) who required an extra source of $HO_2$ in their constrained model of $HO_X$ chemistry in a MBO / terpene emitting environment. Kim et al. (2013) suggested photolysis of OVOCs as the missing source of $HO_X$, or that the $RO_2$ to $HO_2$ conversion is more efficient than modelled. Similar conclusions were drawn by Hens et al. (2014), who suggest that un-measured organic trace gases that are responsible for

"missing" OH reactivity during HUMPPA-COPEC-2010 are a major source of $HO_2$. Hens et al. (2014) also found that between 76 and 90 % of $HO_2$ was formed by reactions of $RO_2$ with NO, depending on the reactivity of OH (lower values at lower OH-reactivity) in agreement with our conclusions that organic peroxy radical reactions with NO were the most important source of $HO_2$ during the campaign.



For OH, we find the main source to be the reaction of $HO_2$ with NO (47 %), while direct formation from $O_3$ photolysis contributes on average only 7 %. As the greatest contribution to $HO_2$ formation is from the reactions of organic peroxy radicals, which are formed mainly via OH + VOCs, this represents efficient $HO_X$ re-cycling even in the low $NO_X$ conditions of the boreal forest. This is in qualitative agreement with (Kim et al., 2013) who found that the reaction of $HO_2$ with NO is ≈

20 times as important as primary production of OH from $O_3$ photolysis in a non-isoprene forested environment with daytime NO levels close to 200 pptv. The results are also consistent with the box-modelling conclusions of Hens et al. (2014) who found the major source of OH to be reaction of $HO_2$ with NO and $O_3$ (73-80 %) with only 20-27% generated directly (i.e. by photolysis of $O_3$, HONO and $H_2O_2$ and reaction of VOCs with $O_3$).

**4 Conclusions**

Simultaneous measurements of PAA, PAN and $H_2O_2$ were used to calculate $HO_X$ levels during the HUMPPA-COPEC-2010 campaign in the boreal forest. A simple expression, based on the measured PAA-to-PAN ratio was used to calculate noon-time $HO_2$ mixing ratios during each campaign day. Good agreement with the model was found on some days, whereas on others the expression resulted in $HO_2$ levels that were a factor of two too high, which is likely related to varying degrees of breakdown of the inherent assumptions of PAA and PAN being in steady state at noon.

A box model, constrained by measurements of relatively long-lived radical precursors such as PAN, $O_3$, NO, $NO_2$, HCHO and $CH_3CHO$ reproduced the observations of PAA, $H_2O_2$ and OH when extra, photolytic sources of $HO_2$ and $CH_3C(O)O_2$ were included (pyruvic acid, glyoxal, methyl-glyoxal and biacetyl). This which was especially important during the biomass-burning influenced period when high levels of many oxygenated VOCs were observed. Measurements of $HO_2$, available for a limited number of days during the campaign, were generally higher (factor of two on average) than those modelled. The

model indicated large concentrations of organic peroxy radicals at nighttime and the apparent discrepancy between modelled and measured $HO_2$ could be resolved by considering the detection of $RO_2$ as $HO_2$ (with an efficiency of 35 %) by the LIF-FAGE instrument during this campaign. Reactions involving organic peroxy radicals form a main $HO_X$ recycling mechanism through the production of $HO_2$ radicals that subsequently react with NO and produce OH. Reactions of acetylperoxy radicals with $HO_2$ and NO were found to be the most important source of $CH_3O_2$ radicals.

An improved data analysis (both analytical and box-model) would require more accurate kinetic data on the reaction of PAA with OH and also on the temperature dependence of PAA formation in the reaction of $CH_3C(O)O_2$ with $HO_2$. Significant uncertainty is also associated with deposition velocities for PAA (in the steady-state analysis) and both PAA and $H_2O_2$ (in the box-model analysis). Future studies of $HO_X$ chemistry in this type of environment would benefit greatly from measurements of vertical gradients (or deposition velocities) of PAA and $H_2O_2$ as well as measurements of speciated $RO_2$,

di-carbonyls and OVOCs.





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



**Acknowledgements**

We thank Uwe Parchatka and Markus Rudolf as well as the technical staff of SMEAR II station for technical support during HUMPPA-COPEC-2010. Funding from Academy of Finland via Center of Excellence in Atmospheric Science and
5    European Commission via ACTRIS-2 is gratefully acknowledged.




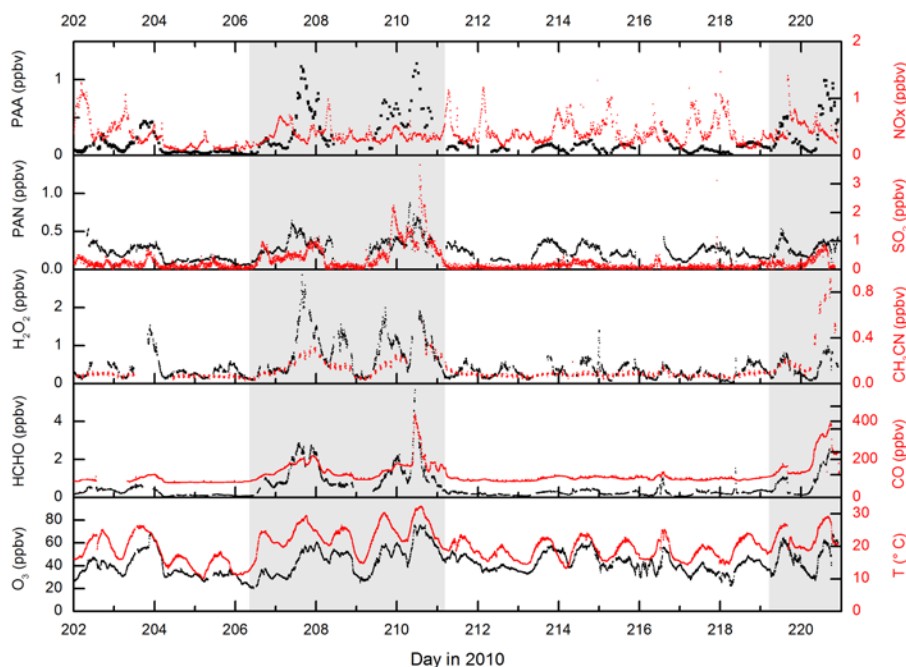

**Figure 1: Measurements of PAA (CH$_3$C(O)OOH), PAN (CH$_3$C(O)O$_2$NO$_2$) and other trace gases during HUMPPA-COPEC-2010. Day 202 was the 20$^{th}$ July, 2010. The shaded areas represent two periods in which the site was impacted by biomass-burning plumes originating from Russia.**





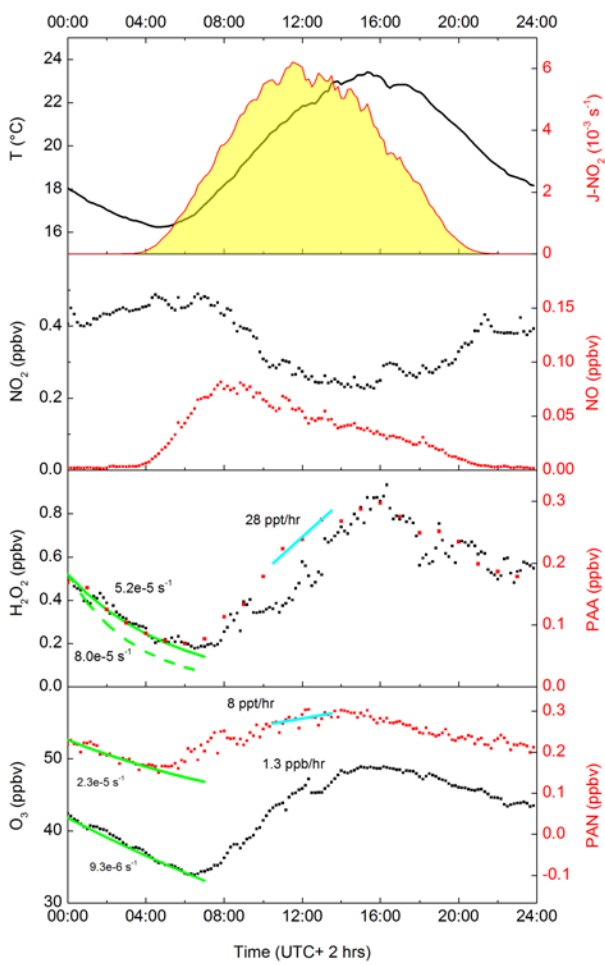

**Figure 2: Averaged diel profiles (day 202-220) of PAN, O$_3$, H$_2$O$_2$, NO$_2$, NO, temperature, J-O($^1$D) (all 10 min averages) and PAA (60 min intervals). The solid green lines represent campaign averaged, nighttime exponential decay rate constants (s$^{-1}$) for H$_2$O$_2$, PAA, PAN and O$_3$. The dashed green line is the adjusted decay constant for H$_2$O$_2$ that takes its formation via ozonolysis of VOCs into account. The cyan lines represent average, production rates (ppbv or pptv per hour) for PAA and PAN (cantered at 12:00).**

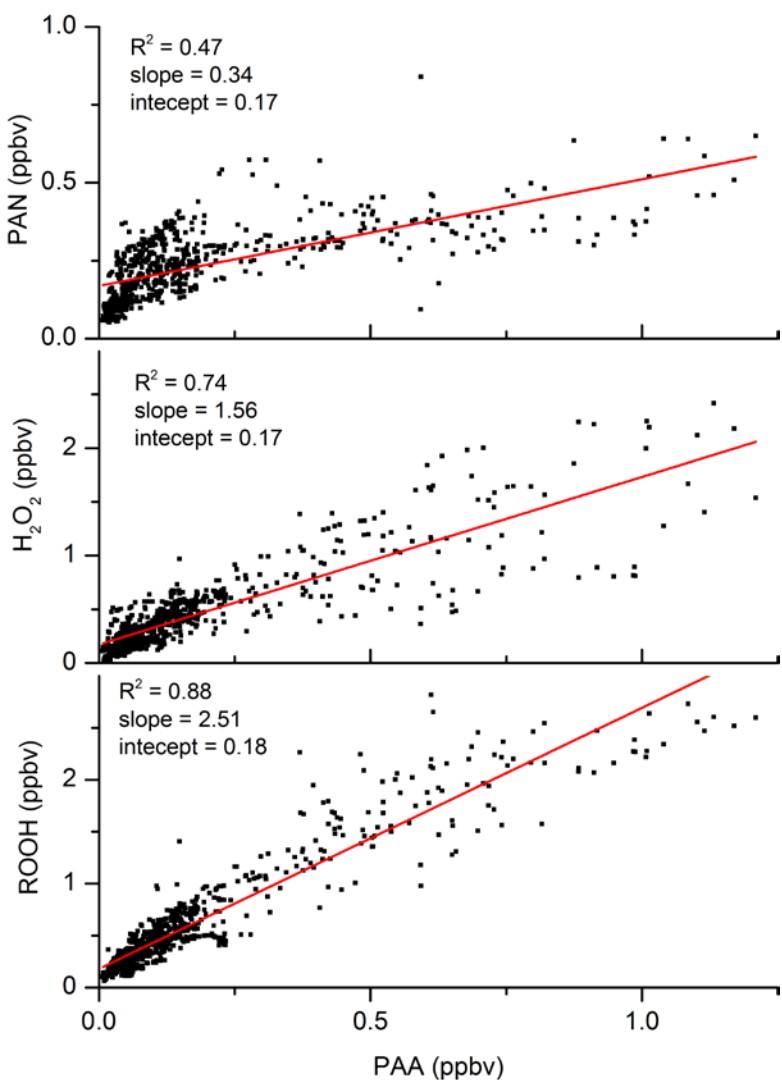

**Figure 3: Correlation of PAA with PAN, H$_2$O$_2$ and ROOH (total organic peroxides).**





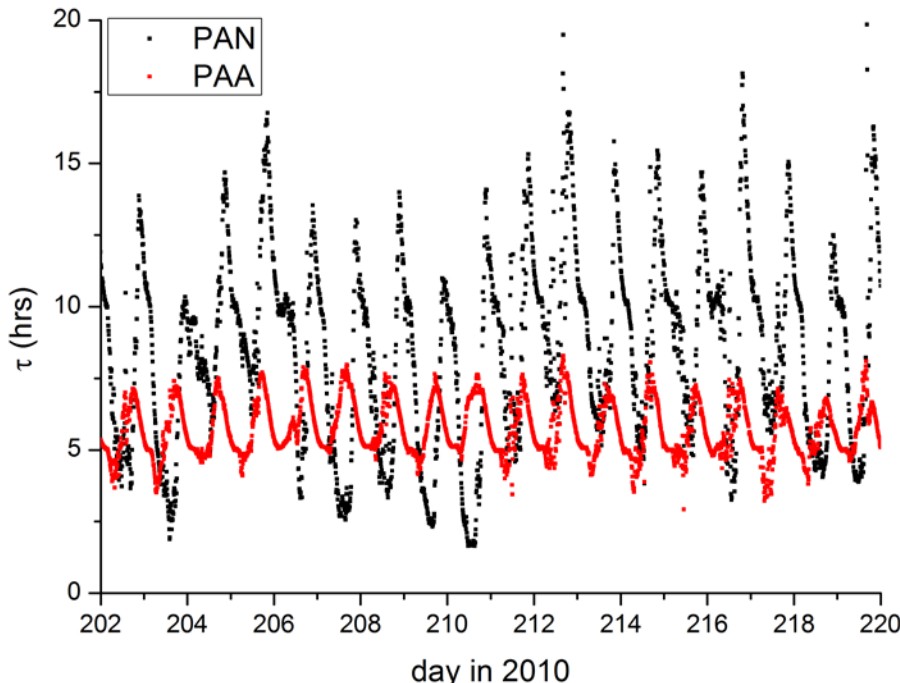

**Figure 4: Lifetimes of PAN and PAA. The more regular diel cycle of PAA reflects the dominance of dry deposition with only a small contribution due to OH (visible as some fine structure in the midday loss rate constant). PAN losses are dominated by thermal decomposition during the day and deposition at night.**



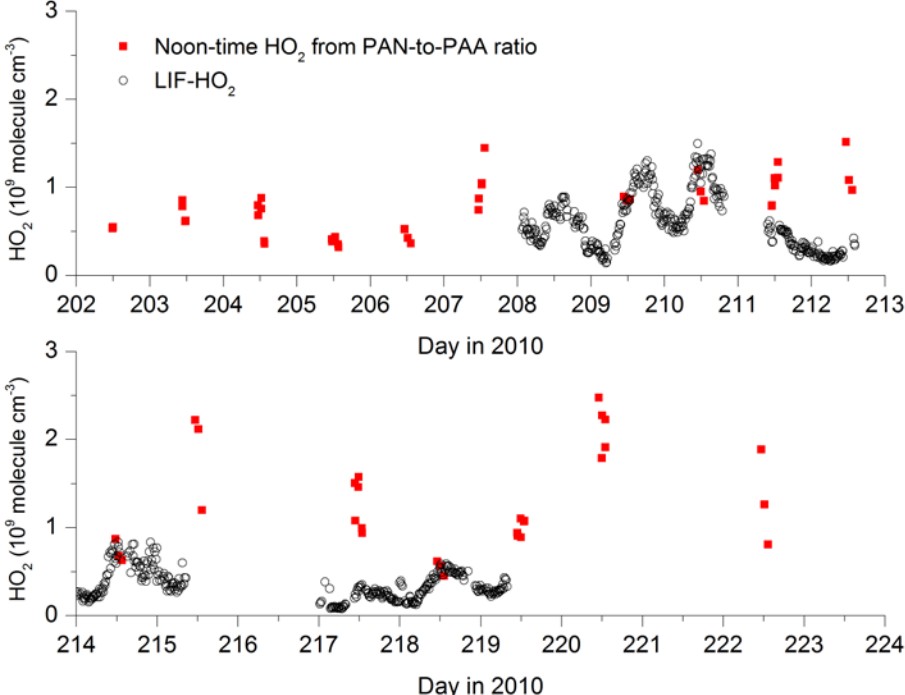

**Figure 5: [HO$_2$] calculated via the PAN-to-PAA ratio at local noon (red symbols) and comparison with measured HO$_2$ (black data-points) by LIF-FAGE.**



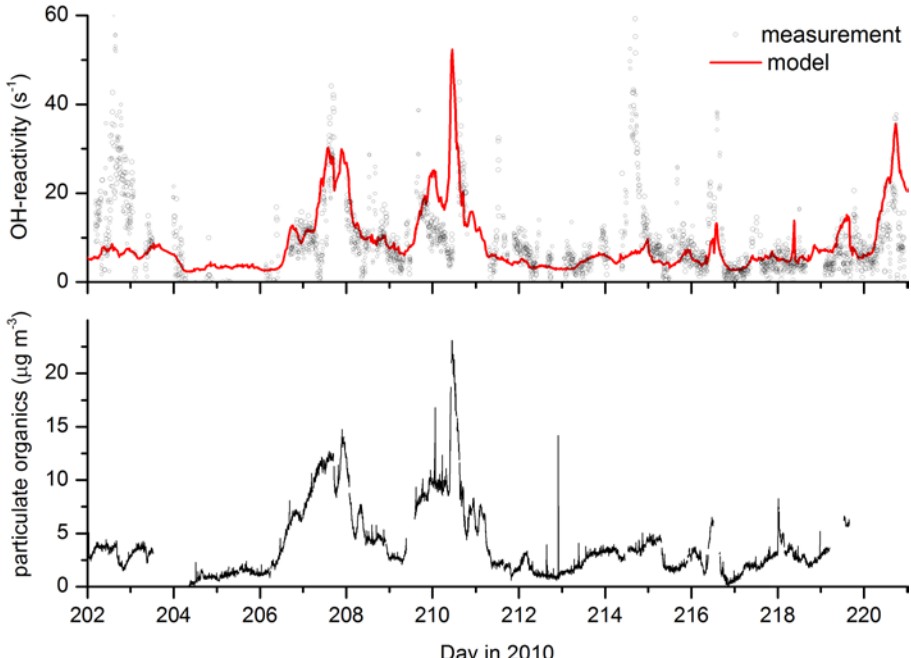

**Figure 6: Upper panel: Measured (black data-points) and modelled OH reactivity (red line). The aerosol organic content (lower panel) shows similar campaign variability to modelled OH-reactivity.**




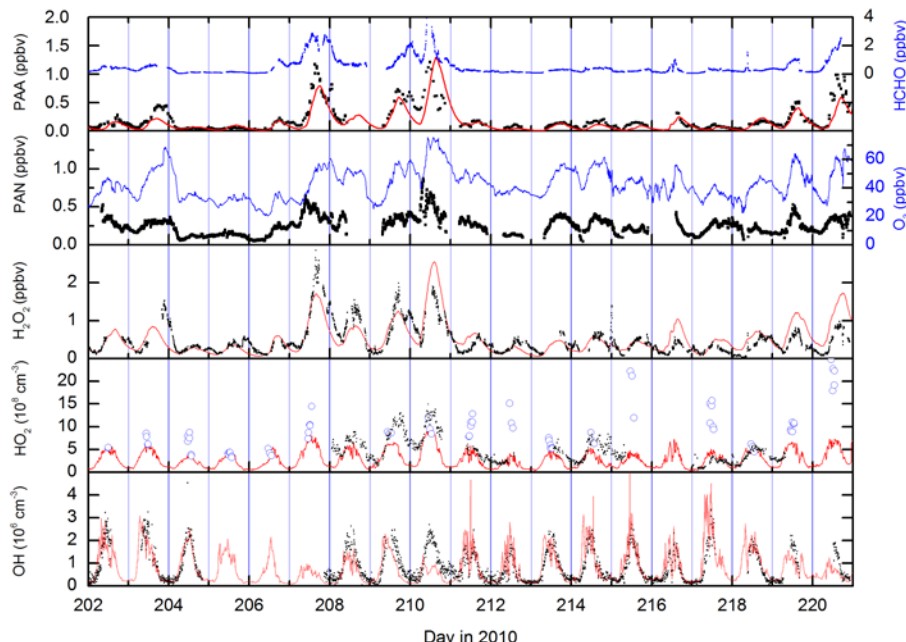

**Figure 7: Measurements (black and violet data-points) and model (red lines). The OH dataset displayed is from CIMS measurements at ground level and has been adjusted to match levels observed at canopy height (see text for details). The open, red circles are HO$_2$ concentrations calculated from the PAN-to-PAA ratio and the steady-state expression (eqn. 4) presented in section 3.1.**





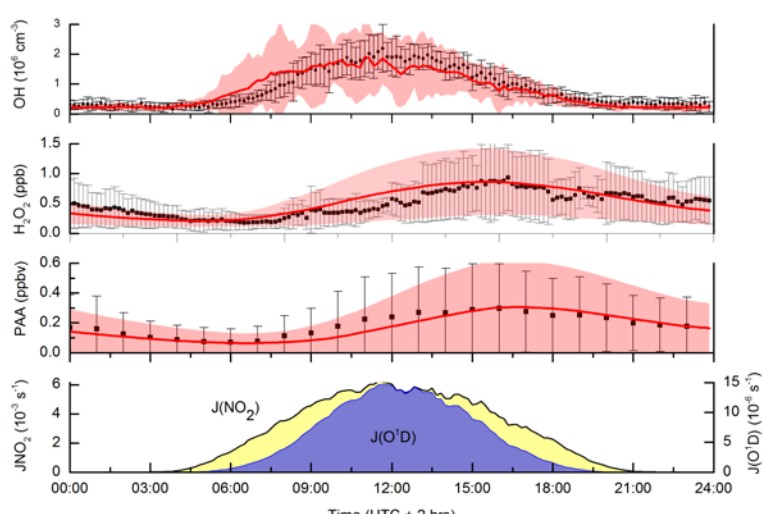

**Figure 8: Campaign averaged (days 202 to 220) diel profiles of OH, PAA and H₂O₂. Measurements are black data-points, error bars are 1 σ variability. Model results are the red lines with the shaded area representing 1 σ variability. The diel averaged photolysis rate constants of NO₂ and O(¹D) are displayed as indicators of photochemical activity.**



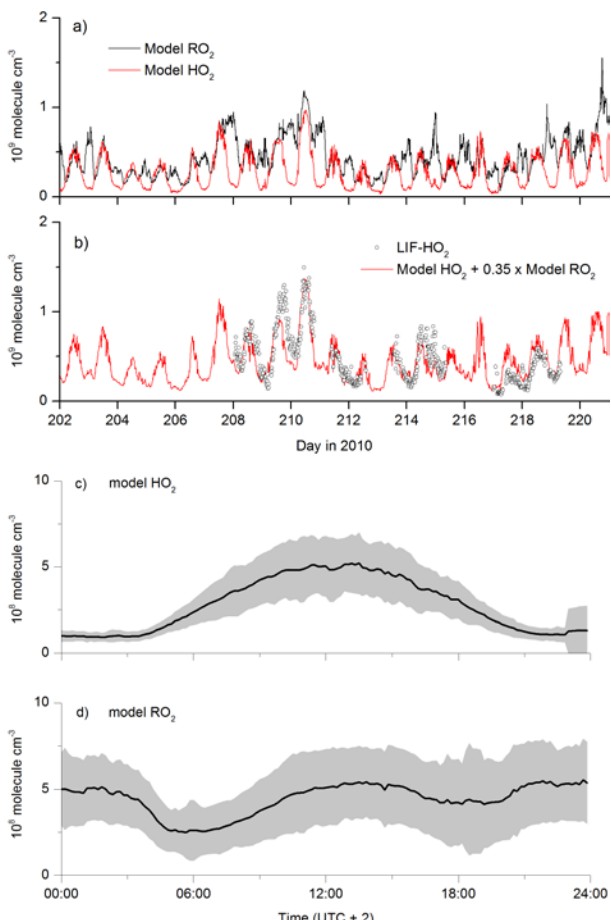

**Figure 9: Modelled HO$_2$ and RO$_2$ concentrations as time series (a) and campaign averaged diel profiles (c and d). RO$_2$\* represents organic peroxy radicals formed in the OH-initiated degradation of OVOCs and terpenes. The red line in plot (b) is the sum of model HO$_2$ concentration plus 0.35 times the model RO$_2$ concentration.**



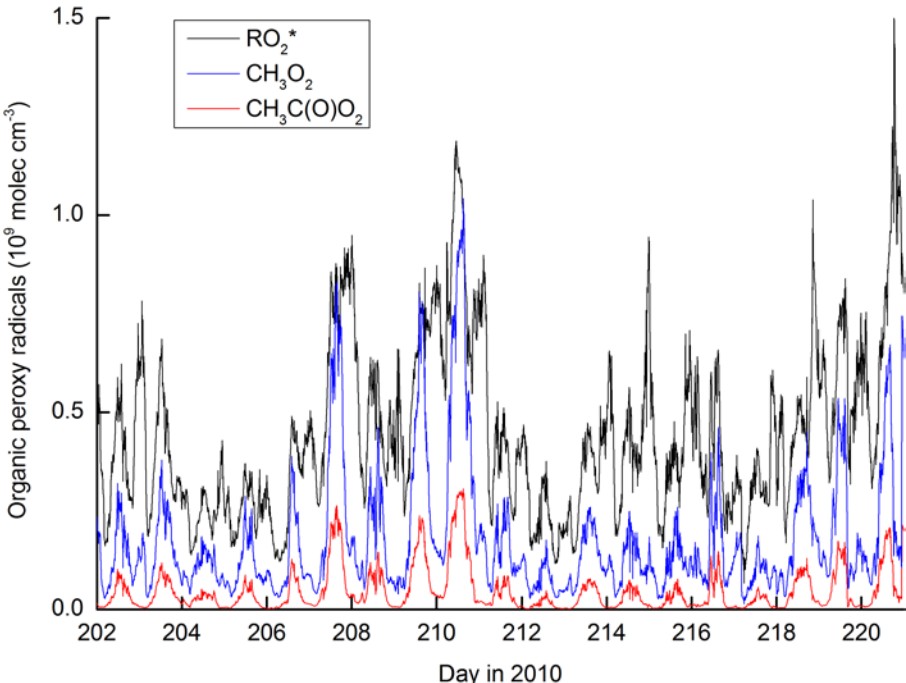

**Figure 10: Modelled organic peroxy radical concentrations. RO$_2$\* represents organic peroxy radicals formed in the OH-initiated degradation of OVOCs and terpenes.**



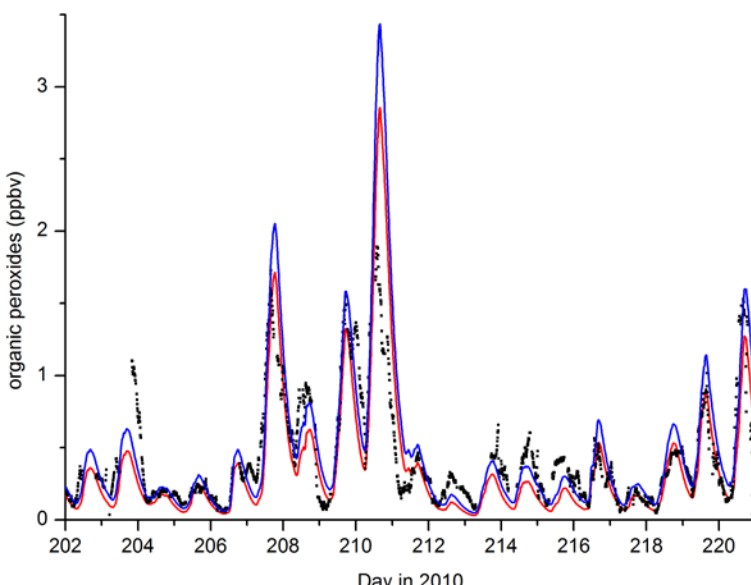

**Figure 11: Total organic peroxide measurement (black data-points) and model results. The red line is the modelled sum of CH₃OOH and PAA, the blue line includes a contribution (10 %) from model ROOH which is formed from RO₂* + HO₂.**

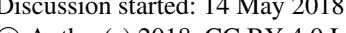



**Figure 12: Measurement versus model results for OH, HO$_2$, (HO$_2$*), H$_2$O$_2$, PAA and ROOH. Red data are daytime, black and grey are dusk and nighttime. The fit lines account for uncertainty in both axes. Uncertainty is statistical (1σ).**



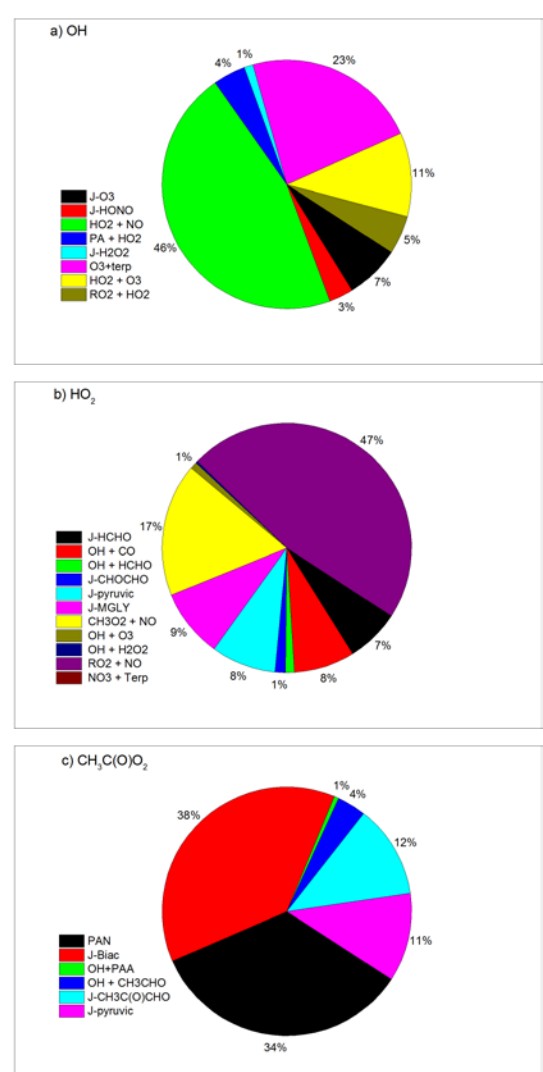

**Figure 13. Modelled relative sources (throughout the diel cycle) of OH, HO$_2$ and CH$_3$C(O)O$_2$ during HUMPPA-COPEC-2010 (days 202-220).**