# Peer review of "Insights into $HO_X$ and $RO_X$ chemistry in the boreal forest via measurement of peroxyacetic acid, peroxyacetic nitric anhydride (PAN) and hydrogen peroxide"

_Atmospheric Chemistry and Physics, 2018_

## Referee Comment (RC1) · Anonymous Referee #2 · 3 Aug 2018

This manuscript describes an extensive set of measurements made in the boreal forest in 2010. The data set includes photochemically active molecules, in addition to OH and HO2 radicals, so that a detailed balance of radical sources and sinks can be assessed. Modifications to a standard model are discussed stepwise, so the reader can follow the logic involved in building the model.

The final agreement between measurements and model is good, with the exception that the HO2 measurements are impacted by other, BVOC-derived peroxy radicals, RO2*, that are partially detected by the instrument. Also, additional photolytic sources

of radicals need to be invoked, e.g., glyoxal, methyl glyoxal and biacetyl, to account for the radical production rates.

Overall, this is a good paper, which is clearly written and explained, and provides a good data set for future comparisons. It can be published after consideration of some minor comments, below. Additionally, there are quite a few grammatical errors (mismatch between subject and verb; misuse of commas) that could be cleaned up by the authors.

Page 1, line 27. "these" peroxy radicals, sounds like it is referring to CH3O2 and CH3C(O)O2, but I think it refers to RO2*?

Page 2, lines 26 and 30. Inconsistency between the values of k2.

Also, in the supplemental table S1, the value of k2 seems to be a factor of 10 lower yet (A-factor should be 3.14E-12).

Page 3, line 4. Mixing ratios and deposition velocities of PAA have also been measured by the CalTech group. See for example Nguyen et al., Proc. Nat. Acad. Sci., 112 (5), E392-E401, 2015. The ambient mixing ratios and fluxes are shown in the Supplemental Appendix, Figure S17.

Page 3, line 26. No need to capitalize "silver".

Page 3, line 31. Probably need to insert "an" before iodide, or say "mass spectrometry" rather than spectrometer.

Page 5, line 1. Insert space between "gases" and first parenthesis.

Page 6, line 31. Refers to Hall and Claiborn, 1997, and Claiborn and Hall, 1997. Are these the same? Presumably the former is correct, according to the reference list.

Page 7, section 3.1. Please check reaction numbers. So on line 13, the decomposition of PAN should be reaction (R-1) not R(2).

Page 7, line 14. (R4) does not involve acetyl peroxy loss.

Page 8, line 6. How important is advection? PAN is fairly long lived, and it is well known that it can be transported over long distances. Would you expect it always to be in a photochemical steady state?

Page 8, last line. Should these PAN loss rates be switched? The maximum value is lower than the night time value.

Page 16. Photolysis rate of biacetyl. Klotz et al. (Int. J. Chem. Kinet., 2001) give j(biacetyl)/j(NO2) = 0.036, measured in the Euphore chamber, roughly a factor of 6 lower than that given here. I realize that the carbonyls introduced here are mostly proxies, but it would be better to use realistic photolysis rates, so that the inferred mixing ratios are also realistic if other people want to reproduce them.

Page 17, line 25. Figure S4 should be Figure S3.

Page 17, last line, has an odd number of parentheses.

Page 18, line 5. Maybe state explicitly that as a result of the internal RO2 isomerizations, the HOM molecules contain –OOH groups, rather than just "highly oxygenated". Also, I am not sure that the two references given are appropriate for HOM formation and loss.

Page 20, top. Again, you should include a comparison with the loss rates for H2O2 and PAA measured by Nguyen et al.

Page 21, line 17. Sentence begins "This which. . ." Not sure what that refers to.

Page 30, last line. "cantered" should be "centered".

Page 35, Figure 7. States that open, red circles are HO2. Should be violet? Also, I maybe missed the section of the text where it talks about adjusting the OH to match levels at the canopy height. Maybe include the section number here, so that it is easier to find.

Supplemental Figure S2. I like this figure. Could maybe include it in the main text?

[Figure]

---

## Referee Comment (RC2) · Anonymous Referee #1 · 10 Aug 2018

Ms.No.: acp-2018-400

The authors describe analysis of a complex data set from a field campaign in the boreal forest in Finland from 2010. They use measurements of CH3C(O)OOH (PAA) and CH3C(O)O2NO2 (PAN, commonly peroxyacetyl nitrate) for a rough estimate on the HO2 radical concentration assuming steady-state behaviour for PAA and PAN within an assumed reaction scheme. For comparison, LIF_HO2 data were available in a limited time range. The agreement is reasonable with exception of episodes where HO2 measurements were strongly influenced by RO2 radicals. Results of a box modelling study support the derived HO2 levels and seem to be in excellent agreement with the measured OH radical concentrations. As a result of this study it can be concluded that the HOx chemistry is well understood.

This manuscript provides a couple of interesting things and meets the criteria for ACP. Some minor points should be considered before final acceptance is recommended:

1. Nothing is said regarding the uncertainty of the PAA and PAN measurements and the consequence for the derived HO2 concentrations. A statement on that is needed. Calculation of the consequential error in HO2 would be fine.
2. There is only very sparse information on the OH measurements, I think it was done by nitrate CIMS with a UHEL instrument. What has been done to "adjust" the ground level OH measurements to canopy level. Please explain. Original data compared with those after adjustment, as shown in fig.7, should be given in SI.
3. Expected RO2* radicals from OH+OVOC/terpenes show concentrations of up to 10(9) cm(-3) and surpass CH3O2 concentrations significantly. That´s a bit surprising for me. Are similar RO2* concentrations known from other forest sites?
4. There are a couple of typos, careful proof-reading is needed.

This manuscript describes an extensive set of measurements made in the boreal forest in 2010. The data set includes photochemically active molecules, in addition to OH and HO2 radicals, so that a detailed balance of radical sources and sinks can be assessed. Modifications to a standard model are discussed stepwise, so the reader can follow the logic involved in building the model.

The final agreement between measurements and model is good, with the exception that the HO2 measurements are impacted by other, BVOC-derived peroxy radicals, RO2*, that are partially detected by the instrument. Also, additional photolytic sources C1 of radicals need to be invoked, e.g., glyoxal, methyl glyoxal and biacetyl, to account for the radical production rates.

Overall, this is a good paper, which is clearly written and explained, and provides a good data set for future comparisons. It can be published after consideration of some minor comments, below. Additionally, there are quite a few grammatical errors (mismatch between subject and verb; misuse of commas) that could be cleaned up by the authors.

---

## Author Comment (AC1) · 20 Aug 2018

The comment was uploaded in the form of a supplement:
https://www.atmos-chem-phys-discuss.net/acp-2018-400/acp-2018-400-AC1-supplement.pdf

---

## Author Comment (AC2) · 20 Aug 2018

The following text contains the reviewer's comments (black), our replies (blue) and the changes made to the manuscript (red).

| Reviewer 1 |
| --- |
| This manuscript describes an extensive set of measurements made in the boreal forest in 2010. The data set includes photochemically active molecules, in addition to OH and HO2 radicals, so that a detailed balance of radical sources and sinks can be assessed. Modifications to a standard model are discussed stepwise, so the reader can follow the logic involved in building the model.
 The final agreement between measurements and model is good, with the exception that the HO2 measurements are impacted by other, BVOC-derived peroxy radicals, RO2*, that are partially detected by the instrument. Also, additional photolytic sources of radicals need to be invoked, e.g., glyoxal, methyl glyoxal and biacetyl, to account for the radical production rates. Overall, this is a good paper, which is clearly written and explained, and provides a  good data set for future comparisons. It can be published after consideration of some minor comments, below. Additionally, there are quite a few grammatical errors (mismatch between subject and verb; misuse of commas) that could be cleaned up by the authors.
 We thank the reviewer for this thorough and positive assessment of the manuscript |
| Page 1, line 27. "these" peroxy radicals, sounds like it is referring to CH3O2 and CH3C(O)O2, but I think it refers to RO2*?
 This is correct. We have modified the manuscript to clarify this and now write:
 "The model indicates that organic peroxy radicals were present at night in high concentrations that sometimes exceeded those predicted for daytime and initially divergent measured and modelled $HO_2$ concentrations and daily concentration profiles are reconciled when organic peroxy radicals are detected (as $HO_2$) at an efficiency of 35 %. Organic peroxy radicals are found to play an important role in the recycling of OH radicals subsequent to their loss via reactions with volatile organic compounds." |
| Page 2, lines 26 and 30. Inconsistency between the values of k2.
 This was an error. We have removed the second (false) listing of k. The text now reads:
 "Under conditions of temperature and pressure found in the lowermost troposphere, the rate coefficients for reaction of $CH_3C(O)O_2$ with $NO_2$ and $HO_2$ are similar ($k_1$ at 298 K and 1 bar pressure is 9.3 x $10^{-12}$ $cm^3$ $molecule^{-1}$ $s^{-1}$) and the relative flux of $CH_3C(O)O_2$ radicals into PAN and PAA formation will depend on the relative abundance of $NO_2$ and $HO_2$." |
| Also, in the supplemental table S1, the value of k2 seems to be a factor of 10 lower yet (A-factor should be 3.14E-12).
 This was a typo.
 We have corrected Table S1 with the correct values. |
| Page 3, line 4. Mixing ratios and deposition velocities of PAA have also been measured by the CalTech group. See for example Nguyen et al., Proc. Nat. Acad. Sci., 112 (5), E392-E401, 2015. The ambient mixing ratios and fluxes are shown in the Supplemental Appendix, Figure S17.
 Nguyen et al are now cited:
 Unlike PAN, there are few measurements of PAA (Fels and Junkermann, 1994; He et al., 2010; Zhang et al., 2010; Nguyen et al., 2015)

 Hall and Claiborn (Hall and Claiborn, 1997) measured deposition rates for summed organic peroxides (mainly $CH_3OOH$) which were a factor of about two-to-three lower than for $H_2O_2$ and Nguyen et al. (2015) measured deposition velocities for PAA over a temperate forest that were a factor two lower than $H_2O_2$. |
| Page 3, line 26. No need to capitalize "silver".
 Correction made |

| |
|---|
| Page 3, line 31. Probably need to insert "an" before iodide, or say "mass spectrometry" rather than spectrometer. |
| Correction made |
| Page 5, line 1. Insert space between "gases" and first parenthesis |
| Correction made |
| Page 6, line 31. Refers to Hall and Claiborn, 1997, and Claiborn and Hall, 1997. Are these the same? Presumably the former is correct, according to the reference list. |
| Yes, these both refer to the same paper. We now write: |
| Nonetheless, the value obtained is entirely consistent with nighttime deposition velocities of $0.8 \pm 0.2$, $1.0 \pm 0.3$ and $1.6 \pm 0.3$ cm s$^{-1}$ reported for $H_2O_2$ deposition over the Canadian boreal forest, with daytime $H_2O_2$ deposition velocities that are a factor 10 ($\pm 5$) larger (Hall and Claiborn, 1997). |
| Page 7, section 3.1. Please check reaction numbers. So on line 13, the decomposition of PAN should be reaction (R-1) not R(2). |
| Correction made |
| Page 7, line 14. (R4) does not involve acetyl peroxy loss |
| R4 has been removed from the list of reactions that remove acetylperoxy. We now write: |
| The relative concentrations of trace gases such as NO, and peroxy radicals which can react with $CH_3C(O)O_2$ (via R2 and R3) to that…. |
| Page 8, line 6. How important is advection? PAN is fairly long lived, and it is well known that it can be transported over long distances. Would you expect it always to be in a photochemical steady state? |
| This question is dealt with in detail in several paragraphs on page 8 and in the discussion and Figure on PAN lifetimes during the campaign. |
| Page 8, last line. Should these PAN loss rates be switched? The maximum value is lower than the night time value. |
| No, the loss rate constants are correct, which is related to changes in the BL-height as described in preceding text. |
| Page 16. Photolysis rate of biacetyl. Klotz et al. (Int. J. Chem. Kinet., 2001) give j(biacetyl)/j(NO2) = 0.036, measured in the Euphore chamber, roughly a factor of 6 lower than that given here. I realize that the carbonyls introduced here are mostly proxies, but it would be better to use realistic photolysis rates, so that the inferred mixing ratios are also realistic if other people want to reproduce them. |
| We agree and have amended the text (in two places) appropriately: |
| the photolysis rates of pyruvic acid, glyoxal, methyl-glyoxal and biacetyl were modelled as factors (J-dicarbonyl / J-NO$_2$) of 0.033, 0.0076, 0.019 and 0.033. We note that large differences in experimental results (see (IUPAC, 2018) for a summary) and in the preferred values of evaluation panels (Burkholder et al., 2015; IUPAC, 2018) indicate that the J-values of the di-carbonyls are associated with significant uncertainties. For biacetyl, the factor used, 0.033 is consistent with observations in an environmental chamber (Klotz et al., 2001) where a value of J-dicarbonyl / J-NO$_2$ = 0.036 was reported. |
| and |
| The ratios of $CH_3CHO$ to methyl-glyoxal, biacetyl and glyoxal were varied to optimise the simulation of the measured PAA and $H_2O_2$ mixing ratios and resulted in maximum concentrations of methyl-glyoxal, glyoxal and biacetyl in the model of approximately 0.75, 0.15 and 1.3 ppbv, respectively, which were present during the peak of the biomass burning impacted episodes (when $CH_3CHO$ levels were largest). |
| Page 17, line 25. Figure S4 should be Figure S3. |
| Correction made |
| Page 17, last line, has an odd number of parentheses. |
| Corrected |
| Page 18, line 5. Maybe state explicitly that as a result of the internal RO2 isomerizations, the |

HOM molecules contain –OOH groups, rather than just "highly oxygenated".

We now write:

…form a highly oxygenated molecule (HOM) containing -OOH groups, that may partition substantially to the particle phase…
* * *
Also, I am not sure that the two references given are appropriate for HOM formation and loss.

This was a bad-citation. We now cite Ehn et al:

…and form a highly oxygenated molecule (HOM), that may partition substantially to the particle phase (Ehn et al., 2012)
* * *
Page 20, top. Again, you should include a comparison with the loss rates for H2O2 and PAA measured by Nguyen et al.

We have now done this and use the results of Nguyen et al. The sentence has been amended to state:

Due to lack of experimental data, there is some uncertainty associated with the physical constants describing the main loss processes for PAA and the preferred value of the rate constant for its reaction with OH carries an uncertainty of a factor of two (IUPAC, 2018). The nighttime loss-rate constants for $H_2O_2$ and PAA were derived from measurements but the daytime values were scaled using literature data for $H_2O_2$ as described above. So far, we have adopted the results of Nguyen et al. (2015) and used a daytime deposition velocity of PAA that is 50% that of $H_{2O2}$
* * *
Page 21, line 17. Sentence begins "This which: : :" Not sure what that refers to.

This was poorly phrased. We now write:

Inclusion of these radical sources was especially important during the biomass-burning influenced period
* * *
Page 30, last line. "cantered" should be "centered".

Correction made
* * *
Page 35, Figure 7. States that open, red circles are HO2. Should be violet? Also, I maybe missed the section of the text where it talks about adjusting the OH to match levels at the canopy height. Maybe include the section number here, so that it is easier to find.

We now provide this information:

The OH dataset displayed is from CIMS measurements at ground level and has been adjusted to match levels observed at canopy height (see supplementary information and Hens et al., 2014).
* * *
Supplemental Figure S2. I like this figure. Could maybe include it in the main text?

The deposition velocities of PAA and $H_2O_2$ are not the central theme of this paper and we prefer to keep the Figure in the Supplementary information.
* * *
Reviewer 2

The authors describe analysis of a complex data set from a field campaign in the boreal forest in Finland from 2010. They use measurements of CH3C(O)OOH (PAA) and CH3C(O)O2NO2 (PAN, commonly peroxyacetyl nitrate) for a rough estimate on the HO2 radical concentration assuming steady-state behaviour for PAA and PAN within an assumed reaction scheme. For comparison, LIF_HO2 data were available in a limited time range. The agreement is reasonable with exception of episodes where HO2 measurements were strongly influenced by RO2 radicals. Results of a box modelling study support the derived HO2 levels and seem to be in excellent agreement with the measured OH radical concentrations. As a result of this study it can be concluded that the HOx chemistry is well understood.

This manuscript provides a couple of interesting things and meets the criteria for ACP. Some minor points should be considered before final acceptance is recommended:

We thank the reviewer for this thorough and positive assessment of the manuscript

1. Nothing is said regarding the uncertainty of the PAA and PAN measurements and the consequence for the derived HO2 concentrations. A statement on that is needed. Calculation of the consequential error in HO2 would be fine.

We now estimate the uncertainty in $[HO_2]$ calculated using equation (4): Uncertainties of 30%, 25% and 15% for the mixing ratios of PAA, PAN and $NO_2$, respectively can be combined with uncertainties of 10-20% for the rate coefficients $k_1$, $k_{2a}$ and $k_{-1}$ and 50% uncertainty for the deposition velocities for PAA and PAN resulting in an overall uncertainty in $[HO_2]$ of close to 75%.

2. There is only very sparse information on the OH measurements, I think it was done by nitrate CIMS with a UHEL instrument. What has been done to "adjust" the ground level OH measurements to canopy level. Please explain. Original data compared with those after adjustment, as shown in fig.7, should be given in SI.

A detailed comparison between the FAGE and CIMS measurements of OH has been given by Hens et al (2014) and the correction procedure is briefly discussed in the supplementary information. We see little benefit in repeating this here.

3. Expected RO2* radicals from OH+OVOC/terpenes show concentrations of up to 10(9) cm(-3) and surpass CH3O2 concentrations significantly. That's a bit surprising for me. Are similar RO2* concentrations known from other forest sites?

$CH_3O_2$ is thought to be the dominant organic peroxy radical in environments that are not directly impacted by biogenic emissions. However, as the OH losses in the boreal environment are dominated by reactions with large biogenic organics it is not surprising that the first generation peroxy radicals generated are not C1 but larger. The very low levels of NO and $NO_3$ measured during the campaign at night allow $RO_2$ to build up at night, the concentration acquired limited e.g. by self-reaction and reaction with $HO_2$.

We are not aware of measurements of $RO_2$ in forested regions that may confirm the dominance of $RO_2*$ over $CH_3O_2$, but note that most peroxy-radical instruments to date have not been able to speciate the $RO_2$ detected. We allude to this in the conclusions where we suggest that measurements of specific $RO_2$ would be useful to confirm these model results.

Future studies of $HO_X$ chemistry in this type of environment would benefit greatly from measurements of vertical gradients (or deposition velocities) of PAA and $H_2O_2$ as well as measurements of speciated $RO_2$, di-carbonyls and OVOCs.

4. There are a couple of typos, careful proof-reading is needed.

We have carefully proof-read the manuscript